



# Aerosols at the Poles: An AeroCom Phase II multi-model evaluation

Maria Sand[1], Bjørn H. Samset[1], Yves Balkanski[2], Susanne Bauer[3], Nicolas Bellouin[4], Terje K. Berntsen[1,5], Huisheng Bian[6], Mian Chin[7], Thomas Diehl[8], Richard Easter[9], Steven J. Ghan[9], Trond Iversen[10], Alf Kirkevåg[10], Jean-François Lamarque[11], Guangxing Lin[9], Xiaohong Liu[12], Gan Luo[14], Gunnar Myhre[1], Twan van Noije[14], Joyce E. Penner[19], Michael Schulz[10], Øyvind Seland[10], Ragnhild B. Skeie[1], Philip Stier[15], Toshihiko Takemura[16], Kostas Tsigaridis[3], Fangqun Yu[13], Kai Zhang[17,9], Hua Zhang[18]

[1]Center for International Climate and Environmental Research – Oslo (CICERO), Oslo, Norway
[2]Laboratoire des Sciences du Climat et de l'Environnement, CEA-CNRS-UVSQ, Gif-sur-Yvette, France
[3]NASA Goddard Institute for Space Studies and Columbia Earth Institute, New York, NY, USA
[4]Department of Meteorology, University of Reading, Reading, UK
[5]Department of Geosciences, University of Oslo, Oslo, Norway
[6]Earth System Science Interdisciplinary Center, University of Maryland, College Park, MD, USA
[7]NASA Goddard Space Flight Center, Greenbelt, MD, USA
[8]Directorate for Sustainable Resources, Joint Research Centre, European Commission, Ispra, Italy
[9]Pacific Northwest National Laboratory, Richland, WA, USA
[10]Norwegian Meteorological Institute, Oslo, Norway
[11]National Center for Atmospheric Research, Boulder, CO, USA
[12]Department of Atmospheric Science, University of Wyoming, USA
[13]Atmospheric Sciences Research Center, State University of New York at Albany, New York, USA
[14]Royal Netherlands Meteorological Institute, De Bilt, The Netherlands
[15]Department of Physics, University of Oxford, Oxford, UK
[16]Research Institute for Applied Mechanics, Kyushu University, Fukuoka, Japan
[17]Max Planck Institute for Meteorology, Hamburg, Germany
[18]Laboratory for Climate Studies, National Climate Center, China Meteorological Administration, Beijing, China
[19] Climate and Space Sciences and Engineering, University of Michigan, Ann Arbor, MI, USA

*Correspondence to*: Maria Sand (maria.sand@cicero.oslo.no)

**Abstract.** Atmospheric aerosols from anthropogenic and natural sources reach the Polar Regions through long-range transport. Such transport is however poorly constrained in present day global climate models, and few multi-model evaluations of Polar anthropogenic aerosol radiative forcing exist. Here we compare the aerosol optical depth (AOD) at 550nm from simulations with 16 global aerosol models from the AeroCom phase II model inter-comparison project with available observations at both Poles. We show that the annual mean multi-model median is representative of the observations in Arctic, but that the inter-model spread is large. We also document the geographical distribution and seasonal cycle of the AOD for the individual aerosol species; black carbon (BC) from fossil fuel and biomass burning, sulfate, organic aerosols (OA), dust and sea-salt. For a subset of models that represent nitrate and secondary organic aerosols (SOA), we document the role of these aerosols at high latitudes.



The seasonal dependence of natural and anthropogenic aerosols differs with natural aerosols peaking in the winter (sea-salt) and spring (dust), whereas AOD from anthropogenic aerosols peaks during late spring/summer. The models produce a median annual mean (AOD) of 0.07 in the Arctic (defined here as north of 60° N). The models also predict a noteworthy aerosol transport to the Antarctic (south of 70° S) with a resulting AOD varying between 0.01-0.02. The models have also estimated

the shortwave anthropogenic radiative forcing contributions to the direct aerosol effect (DAE) associated with BC and OA from fossil fuel and biofuel (FF), sulfate, SOA, nitrate, and biomass burning from BC and OA emissions combined. The Arctic modeled annual mean DAE is slightly negative (-0.12 W m$^{-2}$), dominated by a positive BC FF DAE during spring and a negative sulfate DAE during summer. The Antarctic DAE is governed by BC FF. We perform sensitivity experiments with one of the AeroCom models (GISS modelE) to investigate how regional emissions of BC and sulfate and the lifetime of BC

influence the Arctic and Antarctic AOD. A doubling of emissions in East Asia, result in a 33 % increase in Arctic AOD of BC. However, radical changes such as reducing the e-folding lifetime by half or doubling it, still fall within the AeroCom model range.

## 1 Introduction

The Polar Regions are relatively free from local sources of anthropogenic climate drivers, but are still experiencing rapid

changes to increasing greenhouse gas concentrations that are distributed globally. These changes are amplified by feedbacks in the system, such as temperature feedbacks and the ice albedo feedback (Pithan and Mauritsen, 2014). The temperature in the Arctic is experiencing increases that are twice the global rate, resulting in reductions in summer sea-ice (Hartmann et al., 2013; Screen and Simmonds, 2010). In the Antarctic, summer sea-ice is increasing while several interior regions are rapidly losing ice mass (Rignot et al., 2008). The role of aerosols in the ongoing Polar climate changes is not well understood. Yang

et al. (2014) emphasize the importance of including aerosols in models when simulating the recent changes in the Arctic climate. For instance, the measured decrease in anthropogenic sulfate concentrations in the Arctic during the last decades (Hirdman et al., 2010; Quinn et al., 2009) may have had a warming effect on the Arctic (Navarro et al., 2016). Shindell and Feluvegi (2009) showed that decreasing sulfate and increasing BC concentrations during the last three decades have substantially warmed the Arctic. In general, the climate impacts of aerosols and clouds constitute one of the largest sources of

uncertainty in climate models (Boucher et al., 2013). This is true on a global scale and likely for the critical Polar regions where both sensitivities and dynamical processes may differ significantly from global mean values. Reducing these uncertainties is crucial to improve the reliability of future climate projections.

Aerosols perturb the Earth's radiation balance through extinction of solar radiation (McCormick and Ludwig, 1967; Schulz et al., 2006). By scattering solar radiation, aerosols produce a negative shortwave radiative forcing (DAE) at the top-of-the-

30 atmosphere (TOA). Some aerosols such as BC and dust also absorb solar radiation, and this absorption can lead to a positive





DAE TOA (Bond et al., 2013). For a given aerosol abundance, the magnitude and sign of the DAE depend on the underlying surface albedo (Haywood and Shine, 1995). In the Polar Regions, the high albedo of snow and ice covered surfaces will increase the absorption associated with the DAE for absorbing aerosols (Hansen and Nazarenko, 2004; Bond et al., 2013). Concurrently, deposition of BC and dust can reduce the surface albedo and promote snowmelt (Flanner et al., 2009; Krinner et al., 2006; Clarke and Noone, 1985). Aerosols also influence the energy balance by changing the optical properties and lifetime of clouds (Twomey, 1977; Albrecht, 1989), and through changes to atmospheric stability (Hansen et al., 1997; Hodnebrog et al., 2014; Samset and Myhre, 2015).

The amount of aerosols emitted into the atmosphere has increased over the industrial era. Myhre et al. (2013) reported on the DAE due to anthropogenic aerosols in AeroCom Phase II. The global model-median DAE of the total aerosol effect, taking into account changes to BC, sulfate, OA, biomass burning aerosols, nitrate and secondary organic aerosols, was estimated to -0.27 W m$^{-2}$ with an inter-model range of -0.58 to -0.02 W m$^{-2}$ for the time period 1850-2000. Modifying the results from models with missing SOA and nitrate by use of results from the other models and scaling the period to 1750-2010, resulted in a median DAE of -0.35 W m$^{-2}$.

Most of the aerosols in the Polar Regions originate from lower and mid-latitudes (Koch and Hansen, 2005; Hirdman et al., 2010). Large-scale planetary circulations in the Northern Hemisphere govern the transport into the Arctic. The pronounced seasonal cycle of Arctic AOD typically has a maximum during late winter and spring due to a winter-time build-up in the shallow boundary layer with effective transport and reduced scavenging, often referred to as the Arctic Haze (Iversen and Joranger, 1985; Stohl, 2006). The seasonal cycle of Arctic AOD also varies spatially due to changing emissions, composition and transport patterns. In spring pollution haze and dust plumes from Asian deserts are most common, while later in the season biomass and wildfire smoke from North America and Siberia are observed more frequently (Tomasi et al., 2007). As the Arctic sea-ice is melting and more open water is exposed, emissions of sea-salt, dimethylsulfide (DMS), and organic aerosols within the Arctic are expected to increase (Nilsson et al., 2001; Browse et al., 2014).

In the Antarctic, sea-salt particles dominate the coastal sites, which are strongly influenced by the surrounding ocean (Tomasi et al., 2007). In summer, the sulfate from DMS produced by phytoplankton is at its peak, which can also influence the aerosol distribution in the Antarctic (Arimoto et al., 2004). The stations on the Antarctic Plateau on the other hand are mostly influenced by long-range transport and the subsidence of fine sulfate and methane sulfonic acid (MSA) (Hara et al., 2004; Bigg, 1980). Sea-salt measured here originates mainly from marine air transported by large storm events. The Antarctic is also influenced by smoke aerosols transported from South America and Southern Africa.



The ability of climate models to simulate aerosol burdens in remote regions depends on the transport and precipitation, as well internal aerosol physical and chemical parameterizations, such as wet deposition, oxidation and microphysics (Shindell et al., 2008; Textor et al., 2006; Zhou et al., 2012; von Hardenberg et al. 2012). Aerosol observations in the Polar Regions are sparse. Previous comparisons between models and single observations show significant model biases (Stohl et al., 2013; Shindell et

al., 2008; Koch et al., 2009). Eckhardt et al. (2015) evaluate sulfate and BC concentrations from different models against a large set of ground based and aircraft measurements in the Arctic. They find that the aerosol seasonal cycle at the surface is weak in most models and that the concentrations of equivalent BC and sulfate are underestimated in winter/spring, but improved relative to earlier comparisons. Jiao et al. (2014) compare AeroCom phase II models with observations of BC in snow in the Arctic. They find that simulated BC distributions in snow are not well correlated with measurements, but that

averaged values over the measurement domain are close to observed. The BC atmospheric residence time in the Arctic varies from 3.7 to 23.2 days in the models, and they suggest that aerosol removal processes are a leading source of variation in model performance in the Arctic. Kristiansen et al. (2016) calculate the aerosol lifetime by using observations of two radioactive isotopes released from the Fukushima nuclear power plant accident; one passive tracer and one that condenses on sulfate particles ($^{137}$Cs) that were used as a proxy for sulfate aerosols' fate in the atmosphere. Based on surface measurements taken

in the weeks after the release, they derive an e-folding lifetime of 14.3 days for $^{137}$Cs that serve as an estimate of the lifetime of sulfate. They compare this estimate with 19 AeroCom phase II models initialized with the same identical emissions of $^{137}$Cs and the passive tracer. The AeroCom models show a large spread in their estimates lifetimes (4.8 to 26.7 days) and a mean of 9.4 days, which is low compared to the measurements (14.3 days). The underestimation is larger for the northernmost stations, suggesting that the models remove aerosols too quickly and also underestimate the transport to the Arctic.

Here we present results from phase II of the AeroCom model experiment. The goal is to document the seasonal cycle of mean aerosol abundances and the resulting DAE at the poles, predicted by climate models presently in use, and the multi model spread. The DAE does not include indirect clouds effects or surface albedo modifications. As global aerosol emissions may change rapidly, both in magnitude and geographical distribution, and aerosol abundance observations in the Polar Regions are sparse, one aim of the present study is to deliver a baseline to which future model studies and observations may be compared.

## 25 2 Methods

### 2.1 Models

We have used results from 16 global aerosol models that participated in the AeroCom phase II project (e.g. Myhre et al., 2013) http://aerocom.met.no. The models are NCAR-CAM3.5 (Lamarque et al., 2010; Lamarque et al., 2012), CAM4-Oslo (Kirkevåg et al., 2013), CAM5.1 (Liu et al., 2012), GISS-MATRIX (Bauer et al., 2008), GISS modelE (Koch et al., 2011),



GMI-MERRA-v3 (Bian et al., 2009), GOCART-v4 (Chin et al., 2009), HadGEM2 (Bellouin et al., 2011), IMPACT (Lin et al., 2012), INCA (Szopa et al., 2013), ECHAM5-HAM2 (Stier et al. 2005;Zhang et al., 2012b), OsloCTM2 (Skeie et al., 2011), SPRINTARS (Takemura et al., 2005), TM5 (Vignati et al., 2004), GEOS-Chem (Yu, 2011), and BCC (Zhang et al., 2012a). Model descriptions including model resolution, dynamics, and microphysics schemes used are given in Table 1 and 2 in Myhre et al. (2013).

Each model has provided climate and aerosol simulations using year 2006 meteorology. For present-day simulations emissions for year 2000 have been used, and for preindustrial runs year 1850 emissions have been used (Lamarque et al., 2010). All AeroCom models include sulfate, BC, primary organic carbon, sea salt, and mineral dust in their total AOD, and some models also include nitrate and SOA. To report on the individual species, the models have either added double calls in the radiation code or performed additional runs where each specie has been run with preindustrial emissions. However, not all models were able to extract the AOD for the individual species. Table 1 lists the models and the species reported by each model. The individual species include BC (from fossil fuel, biofuel and biomass emissions), sulfate, total OA (from fossil fuel, biofuel and biomass emissions), nitrate, SOA, sea-salt, and dust. Here, organic aerosols refer to the total mass of organic compounds in the aerosol (both primary and secondary). For a comprehensive documentation on OA and SOA treatment in the AeroCom Phase II models, see Tsigaridis et al. (2014). The AOD is a measure of the total extinction (scattering + absorption) of sunlight as it passes through the atmosphere. In this study we use the AOD at 550nm wavelength. The models have estimated AOD as a combination of aerosol abundances and optical properties, which is why AOD can be reported in the months where there is no actual sunlight. The DAE is calculated as the difference in TOA SW radiation between simulations with present-day and preindustrial emissions of aerosols and their precursors. Results are available for total aerosol forcing, as well as for individual aerosol species (BC from fossil fuel and bio fuel emissions (FF), sulfate, total OA FF, nitrate, SOA, and OA and BC combined from biomass burning (BB) emissions. Hereafter we will use the term 'BC' for total BC (from fossil fuel, biofuel and biomass emissions) and BC FF for anthropogenic BC (from fossil fuel and biofuel emissions), and the same is used for total OA. For AOD we report BC (and OA) only, and for DAE we distinguish between BC FF, OA FF and biomass, which latter consists of emissions from both BC and OA. For information on the radiative transfer schemes of the individual models, see Stier et al. (2013), their Table 2, and the aerosol model references in Myhre et al. (2013), their Table 2.

Even if the models used meteorology for the year 2006, there is some inter-model variability in the simulated winds. Some models are nudged to different sets of reanalysis while others have used different prescribed meteorology data sets, see Table 1 in Myhre et al. (2013). Three models (NCAR-CAM3.5, CAM4-Oslo, and CAM5.1) have calculated the meteorology online, i.e. with free-running meteorological fields. In CAM4-Oslo the meteorology is calculated based on the CAM4 aerosol extinction and cloud droplet fields, which does not differ between preindustrial and present-day simulations. The two other models have been run for several years to account for the year-to-year variability, and the reported simulations are based on a 5-year average. The fields (winds, temperature, humidity) are not identical between the preindustrial simulations and the



present-day simulations in these two models. The calculated aerosol-induced climate response in the polar regions will therefore be due to a combination of differences in emissions and in transport and lifetime. One model (GISS modelE) has duplicate 6-year runs with both nudged winds and free-running winds for preindustrial and present-day conditions, and we find that the difference in the Arctic fraction of the transported tracers between the nudged and the free wind simulations is small. The difference varies between 0.1-1.0 % for most species (up to 2.0 % for a few species), for both preindustrial and present-day simulations. Another study with the CAM5.3 MAM4 model finds a significant difference in BC concentrations on a global scale between nudged and free-running winds (Liu et al. 2016), while the differences between the nudged and un-nudged runs in a study with ECHAM-HAM were small (von Hardenberg et al. 2012). Nevertheless, it did not make much difference for the ensemble results whether we included or excluded the three models that generated their own winds, and we have therefore decided to include these models in the analysis.

There is no unique definition of the Arctic region and here we have defined the Arctic as the region north of 60° N, a definition found in other studies (Shindell and Faluvegi, 2009; AMAP, 2011). To avoid a large influence from the Southern Ocean we have defined the Antarctic as the region south of 70° S.

### 2.2 Comparison with observations

### 2.2.1 AERONET

We have compared the modeled seasonal cycle of AOD in the grid-box of each model in which the respective station is located with ground-based measurements from twelve stations in the Arctic and Antarctic from the AErosol RObotic NETwork (AERONET) http://aeronet.gsfc.nasa.gov/ (Holben et al., 1998). The coordinates and measurement years for each station are given in Table 2. For Barrow, Alert, and Ny-Ålesund, we have used monthly mean climatology derived from daily mean of spectral AOD reported in Stone et al. (2014). For the three stations in Antarctica, one coastal (Neumayer), one mid-altitude (Troll), and one Plateau (South Pole) monthly mean climatology is taken from Tomasi et al. (2015).

### 2.2.2 MODIS

Due to the high reflectance over bright surfaces, obtaining reliable satellite retrieval of AOD is difficult at the Poles. Glantz et al. (2014) compared AOD 555nm from the Moderate Resolution Imaging Spectroradiometer (MODIS) Aqua collection 5 Level 2 (Remer et al., 2005) over (dark) ocean areas around Svalbard with available AERONET ground-based measurements at Svalbard (Longyearbyen (78.2°N, 15.6°E) and Hornsund (77.0°N, 15.6°E)) for the period 2003-2011. They found comparable values in the summer season (JJA) (0.041 ± 0.025 for MODIS and 0.043 ± 0.024 for AERONET) and early autumn (September) (0.035 ±0.021 for MODIS and 0.038 ±0.021 for AERONET), but larger differences during spring (0.115 ±0.069 for MODIS and 0.093 ±0.050 for AERONET). The spring differences are partly explained by diverse air masses causing



inhomogeneous aerosol geographical distributions. Glantz et al. (2014) conclude that satellite AOD retrievals in the Arctic marine atmosphere varies within the expected uncertainties of MODIS retrieval over ocean and can be of use to climate model validation. We have compared the MODIS AOD values with the AeroCom models averaged over the same area (75° N-82° N, 10° W- 40° E). For details on the retrieval, see Glantz et al. (2014).

## 2.2.3 CALIOP

We have compared modeled AOD with retrieved AOD from the Cloud-Aerosol LIdar with Orthogonal Polarization (CALIOP) onboard the Cloud–Aerosol Lidar and Infrared Pathfinder Satellite Observations (CALIPSO) satellite. CALIOP is an active nadir-looking backscattering lidar. It distinguishes clouds and aerosols by using the total backscatter radiation measured at 1063 nm combined with the linear depolarization at 532 nm (Liu et al. 2009). Because it is an active instrument CALIOP can

retrieve aerosol and cloud vertical profiles during day and night, and can measure over the highly reflective surfaces in the Arctic. However, daytime retrievals are affected by the noise from scattering of solar radiation and are therefore less accurate than night-time retrievals (Winker et al., 2009). In the Arctic, there are no night-time observations in May, June and July which complicates the interpretation of spring and summer retrievals. CALIOP reports AOD by integrating the aerosol extinction coefficient from all detected layers over a given location. Thin aerosol layers in the Arctic often have backscattering values

below the detection threshold of CALIOP and the column AOD can therefore be underestimated (Rogers et al. 2014). Omar et al. (2013) found that retrieved AOD from AERONET stations was 25 % higher compared to CALIOP AOD for AOD less than 1. CALIOP has an inclination angle of about 98.14° and has therefore no data points above 82 °N.

## 3 Results

Here we present AOD and DAE results from the Arctic (defined as 60° N - 90° N) and the Antarctic (70° S - 90° S) regions.

We compare the simulated seasonal AOD to ground-based measurements from a selection of stations in the Arctic and Antarctic. We also compare the modelled AOD with retrieval from MODIS over the Svalbard ocean region and with CALIOP. We then document the model simulated regional patterns for each aerosol specie.

### 3.1 Aerosol Optical Depth

Figure 1 shows the seasonal cycle of the total AOD in the Arctic and the Antarctic, for all of the AeroCom phase II models.

Values are for present day conditions, i.e. emissions representative of year 2000. The model-median AOD (shown in thick black line) has a summer maximum and a winter minimum at both Poles, but there is a large variation among the different models. For the Arctic, the spread is larger during the winter/early spring and smaller during the summer months. A few models suggest an earlier Arctic AOD maximum in winter (IMPACT) and early spring (GEOS-Chem and GOCART). For GEOS-



Chem and GOCART this maximum is dominated by natural aerosols (sea-salt and dust, respectively, as shown in Fig. 8). Note that modeled AOD is calculated from simulated aerosol distributions, and can therefore be reported even for months where there is no actual sunlight.

### 3.1.1 Comparisons with measurements at both Poles

We have compared the seasonal cycle of modeled AOD with measurements from nine Arctic stations (details in Table 2), shown in Fig. 2. For many stations, most models fail to capture the observed high spring AOD, especially at Barrow, Alert, Yakutsk and Resolute Bay. There is a better agreement during the summer season when the observed AOD and its variability is lower, but the model median is lower than the observed mean at all stations except Andenes. Bonanza Creek experienced unusually high August values in 2004, 2005 and 2009, resulting in a large standard deviation. The correlation coefficient

between the AeroCom monthly mean and the AERONET monthly mean (Feb - Nov) averaged over the 9 stations is 0.71 ($P < 0.05$).

Figure 3 shows the spring (MAM) and summer (JJA) AOD for each model averaged over the nine Arctic stations, together with the measured AERONET AOD. As is apparent from the correlation coefficient (0.71) and the plots, the multi-model average is not a bad representation of the observed AOD, but the models altogether vary by a factor of 5-6 in magnitude.

HadGEM2, OsloCTM2 and SPRINTARS are the models closest to these observations during summer, and GISS modelE and OsloCTM2 are the models closest during spring.

Retrieval of AOD from the MODIS satellite directly over snow and sea-ice are not available, due to the highly reflectivity of these surfaces. Glantz et al. (2014) have provided spatial averages of MODIS AOD555nm over (darker) ocean areas around Svalbard over a 9-year period, see Methods. In our comparison, we have included this 9-year average to take into account the

20 interannual spread in the data, even though the AeroCom models have simulated one year only. Figure 4 a) shows AOD over the Arctic Ocean (75° N - 82° N, 10° W - 40° E) from MODIS retrieval 2003-2011 from Glantz et al. (2014) compared with the AeroCom models from April through September. The retrieved AOD is approximately 0.1 in spring, but the uncertainty range is large ($0.115 \pm 0.069$ for MODIS and $0.093 \pm 0.050$ for AERONET). The retrieved AOD decreases over summer through September. The AeroCom model mean also show a decrease throughout the year, but the slope is not as steep compared

to MODIS. Some of the models shown in Fig. 4 b) does have a steeper slope (GOCART, GISS-MATRIX, and GEOS-Chem). In the MODIS data the influence from large forest fires events and volcanic eruptions during summer has been removed to represent background conditions, and this might be part of the reason why the models show higher values during summer compared to MODIS.

We have compared total AOD with the vertical integral of the monthly mean elastic backscatter at 532 nm from CALIOP for the years 2006-2007. Figure 5 shows the seasonal cycle of AOD532nm retrieval from CALIPSO for the years 2006-2011



averaged over 60° N - 82° N compared to the AeroCom models screened by CALIOP availability (which is why June has zero numbers north of 60° N). For all months except September, the model median lies within the range of the CALIOP retrievals. The correlation coefficient between the model median monthly mean and the CALIOP monthly mean is 0.75.

Figure 6 shows the measured seasonal cycle of AOD from three Antarctic stations (Neumayer, Troll and South Pole) compared
to the AeroCom models. The root-mean-square-error is shown for each site. We have calculated the root-mean-square-error (rmse) as the square root of the average of the difference between the model median and AERONET values for each month. The two near-coastal sites (Neumayer and Troll) have lower rmse compared to the South Pole station, but with slightly higher spread at Neumayer. Most models seem to be in the lower range of the observations. Tomasi et al. (2015) report multi-year sets of ground-based sun-photometer measurements conducted at nine Antarctic sites. For the high-altitude sites on the
Antarctic Plateau (Dome Concordia 75° S and South Pole 90° S) AOD is very stable, mainly ranging from 0.02-0.04. These values are slightly higher than the median AeroCom Phase II AOD (0.01).

Figure 7 shows a boxplot of the annual mean AOD in the Arctic and Antarctic for total aerosols and for the individual components (sulfate, BC from all sources, OA from all sources, SOA, nitrate, sea-salt, and dust). Model-median total AOD is 0.07 in the Arctic (with a model range of 0.02-0.2), with the largest contribution to Arctic AOD from sulfate (45 %). In the
15 Antarctic the total AOD is 0.01 (0.001-0.05) with sulfate being the largest model median component. However, sea salt shows a large range with the 75 percentile and the maximum value, much higher than the corresponding values for sulfate. Note that all models have reported total AOD, but not all of the individual components' AOD, see Table 1. The AOD median and 25[th]/75[th] percentiles values are listed in Table 3 and Table 4.

Figure 8 shows the seasonal cycle of the AOD for the same individual components as in Fig. 7; sulfate, BC, OA, SOA, nitrate,
sea-salt, and dust averaged over the Arctic region (60° N - 90° N). For sulfate and BC there is a large spread between the models. Most models show a peak in the AOD during summer, with a few models showing a late spring maximum. The geographical distribution over the same region for the summer and winter season is shown in Fig. 9. For sulfate the largest AOD values during summer are found in the Russian and northern Europe regions, while for BC the highest AOD values are found over Russia and East Asia. Both OA and SOA show a maximum during summer in the fire season with the highest AOD
over Russia and East Asia. Summer is also the season with maximum chemical production. During winter the AOD values are low for OA and SOA. There is one outlier for OA, CAM4-Oslo, which has very high marine primary OA emissions, and is the only model that includes MSA in the primary organics emissions (Tsigaridis et al. 2014). The emissions of aerosols (per mass) are dominated by sea-salt and dust. Since these emissions are mostly interactive (a function of wind speed, and also soil moisture for dust), a large model diversity in AOD is not surprising. Sea-salt AOD is highest during the winter season, with a
maximum over the North Atlantic region. The areas around the Norwegian Sea and Barents Sea have the highest cyclonic activity during winter (Serreze and Barrett, 2008). For dust aerosols the models show a maximum in spring/early summer and a secondary maximum in September. The spring maximum originates most likely from dust storms in the Gobi and Taklimakan



deserts, while the second smaller maximum in September might be due to local sources (Barrie and Barrie, 1990). GOCART shows higher AOD values for dust compared to the other models, probably linked to an overestimation of dust emissions (Kim et al., 2014). Only four models have reported AOD from nitrate. The nitrate maximum in winter is located over Eurasia, and over East Asia during summer.

Figure 10 shows the seasonal cycle of AOD in the Antarctic for all the aerosol species. The model-median AOD has a maximum during the SH summer season of 0.02 and is reduced to about half during the winter season (0.01). Modeled BC, sulfate and dust concentrations are highest during the winter months (SH summer).

## 3.2 Aerosol direct radiative forcing

The DAE is calculated as the difference between the reflected solar radiation at TOA between simulations with present day
(2000) and pre-industrial (1850) emissions of anthropogenic aerosols and precursors. Figure 11 shows the multi-model DAE for sulfate, BC FF (from fossil fuel and biofuel emissions), OA FF (from fossil fuel and biofuel emissions), BB, SOA, nitrate, and the total, averaged in the Arctic and the Antarctic regions. In the Arctic the dominant aerosol forcing agents are BC FF and sulfate with model-median DAE estimated at +0.20 and -0.27 W m$^{-2}$, respectively, although the Arctic AOD of BC sis low compared to the other aerosols (Fig. 7). The other treated species are relatively low both in burden and in modeled DAE.
The Arctic annual mean multi-model-median DAE is -0.12 W m$^{-2}$ (with the 25th, 75th percentile; -0.22, 0.03). The Antarctic model-median DAE is 0.02 W m$^{-2}$ (0.01, 0.07). The two largest individual forcing components here are BC FF and OA FF. Note that only the direct radiative forcing is reported. The numbers are listed in Table 5 and Table 6.

Figure 12 shows the Arctic DAE seasonal cycle. The direct influence of aerosols on the radiation budget in the Arctic shifts from a BC-driven positive DAE during spring months, to a sulfate-driven negative forcing during late summer, caused by
higher surface albedo from sea-ice and snow in the former season (Ødemark et al., 2012). Also shown in Fig. 12 is the geographical distribution of DAE during summer (JJA), and a balance between sulfate and BC FF is also apparent here linked to albedo. Negative DAE from sulfate dominates land areas outside the high-Arctic, while higher positive DAE from BC FF is evident in the high-Arctic and in the Pacific.

Even though the AOD of BC is low in the Arctic, the DAE from BC FF dominates the total DAE in spring. In Fig. 13 we have
normalized the JJA DAE (for total, sulfate, and BC FF) to AOD (total, sulfate and BC) to illustrate this. The total normalized forcing is positive in the high-Arctic due to the high efficiency of the BC forcing. Outside the high-Arctic, there is a band of negative direct forcing due to sulfate.

Figure 14 shows the multi-model DAE in the Arctic, sorted by highest-to-lowest, for total aerosol and for sulfate and BC FF. Most models have an annual mean negative net DAE in the Arctic, ranging from -0.3 to 0.0 W m$^{-2}$, while 4 models show a
positive net DAE (HadGEM2, OsloCTM2, GEOS-Chem, and CAM5.1). These latter models have a lower-than-average negative sulfate forcing (HadGEM2, CAM5.1, OsloCTM2), and/or higher-than-average positive BC FF forcing (GEOS-Chem,



OsloCTM2, and HadGEM2). SPRINTARS is one of the models closest to the AOD observations during summer and spring (Fig. 3) and also close to the annual mean model mean DAE. When normalizing the Arctic DAE with AOD for each model (Fig. 15), it is apparent that some models have a high forcing efficiency for sulfate (ECHAM5-HAM2) and/or BC FF (BCC, ECHAM5-HAM2).

5   Figure 16 shows the seasonal cycle of DAE for total aerosols and for sulfate, BC FF, OA FF, BB, SOA, and nitrate in Antarctica. There is a large spread in BC FF DAE during SH summer season. Several of the models that have the highest (positive) BC FF DAE also have the highest (negative) sulfate DAE, as indicated by Myhre et al. (2013). As these models do not show particularly strong forcing per unit AOD (see e.g. Figure 15), but generally have high values for Antarctic AOD (Figure 10), we attribute this correlation to efficient transport of aerosols to the Southern Polar region.

## 10  3.3 Sensitivity simulations with GISS modelE

The model-spread for aerosols at the Poles is large and not entirely surprising, given the large sensitivity to remote transport for aerosol concentrations at high-latitudes. The reasons for this spread include transport and removal mechanisms and the interaction between them. To illustrate some of the variation we have performed sensitivity tests with one of the AeroCom models; GISS modelE. The anthropogenic BC emissions (from fossil fuel and biofuel) have been doubled in South Asia, East

15  Asia and Russia, and Fig. 17a shows the resulting (total) BC AOD in the Arctic for the different regional emission perturbations. The annual mean BC AOD in the Arctic increases by 33 % from a doubling of the BC FF emissions in East Asia, while doubling in South Asia and Russia increase the BC AOD by 10 % and 8 %, respectively. The change in AOD from a doubling in emissions is still within the AeroCom model range, shown in grey lines. Here we only show plots for the Arctic, but the increase in BC AOD in Antarctica is not negligible; 28 % increase for a doubling in East Asia and 7 % for a doubling

20  in South Asia (zero for Russia) (Supplement Fig S1). We have also tested the sensitivity to the e-folding time of BC from hydrophobic (fresh) to hydrophilic (aged) state. Figure 17b shows the resulting change in BC AOD in Arctic by 1) doubling the e-folding time and 2) reducing it by 50 %. By reducing the e-folding time by half, BC decrease by 30 % at both Poles. On the other hand, by making the lifetime longer by doubling the e-folding time, the BC AOD increase with 36-39 % at both poles. The change in BC is still within the AeroCom model range.

## 25  4 Summary and discussion

Recent aerosol-climate models transport anthropogenic aerosols to both Poles, with significant resulting impacts on the local radiative balance. We have reported on modeled AOD and DAE at both Poles, and compared individual and multi-model results to available measurements. Defining the Arctic as the 60° N - 90° N region, the dominant aerosol species, in terms of AOD, are sulfate, sea-salt and OA. The total model-median AOD is 0.07, which is close to observed AOD. However, the inter-



model spread is very wide (0.02-0.2). Compared to measurements at nine Arctic stations, most models tend to underestimate the AOD, especially the build-up of aerosols during early winter/spring. Seasonally, the influence of aerosols on the Arctic energy balance shifts from a BC-driven positive DAE during spring months, to a sulfate-driven negative DAE during late summer. Despite a relatively low Arctic BC AOD compared to the other aerosols, the BC FF DAE dominates during spring

with an annual mean model-median of 0.20 W m$^{-2}$ (0.11, 0.28 W m$^{-2}$ 25$^{th}$/75$^{th}$ percentiles). The total Arctic annual mean DAE model-median is slightly negative, -0.12 W m$^{-2}$ (-0.22, 0.03). We note, however, that this estimate of Arctic aerosol radiative forcing does not include semi- or indirect cloud effects, or surface albedo modification.

The models also predict a fair amount of aerosol transport to the Antarctic region (defined here as 70° S - 90° S). In the Antarctica, modeled AOD is smaller in magnitude than in the Arctic, with an annual mean of 0.01 (0.001-0.05 model range).

Compared to limited available measurements, these values might be on the lower end of the spectrum. As in the Arctic, the dominant aerosol specie is sulfate. The dominant aerosol forcing agent in the Antarctic however is BC, resulting in a small, but positive DAE in this region (0.03 W m$^{-2}$). Again, this does not include possible additional effects of surface albedo modification (Jiao et al. 2014).

Not surprisingly, the spread in modeled AOD at both Poles is large. Interestingly, the spread in modeled DAE is smaller.

Sensitivity experiments of BC with one of the AeroCom models reveal that the Arctic BC AOD is sensitive to the emissions and lifetime of BC. A doubling of fossil fuel and biofuel emissions in East Asia, result in a 33 % increase in Arctic BC AOD. However, radical changes such as reducing the e-folding lifetime by half or doubling it, still fall within the AeroCom model range.

The AeroCom data is only available as monthly averages, so we have therefore compared with monthly averaged retrievals

from AERONET and CALIOP/MODIS. Schutgens et al. (2016a,b) suggests that models should be temporally collocated to the observations before comparing the data to prevent sampling errors. In their study three global models were compared to AERONET/MODIS and sampling errors up to 100% in AOD were apparent for yearly and monthly averages. Since the AeroCom data is only provided monthly, this is a potential problem both for this and most other AeroCom studies.

Various factors lie behind the large spread in modelled AOD in the Polar regions. Recommendations to improve our

understanding of the role of aerosols in the Polar regions and to reduce the uncertainties include sensitivity tests on removal processes (Wang et al., 2013; Liu et al., 2011; Bourgeois and Bey, 2011) and resolution (Ma et al. 2014) during transport to the Arctic, up to date treatment of aerosol mixtures, missing emission sources (Stohl et al. 2013, Evangeliou et al. 2016), and a better characterization of measurement uncertainties in satellite data over Polar land and oceans. Of these, updated emission inventories (global and Polar), and model validation of AODs and column loadings against local observations seem most

pressing to provide a solid baseline for evaluations of transport schemes and calculations of radiative forcing, taking into account a broader range of physical effects.




## 5 Data Availability

AeroCom data and CALIOP data are available through http://aerocom.met.no. Sensitivity studies and further analysis results are available upon request to M. Sand. AERONET data are available at http://aeronet.gsfc.nasa.gov/.

## Acknowledgements

MS has received funding from The Research Council of Norway (RCN) through a FRIPRO Mobility Grant, BlackArc, contract no 240921. The FRIPRO Mobility grant scheme (FRICON) is co-funded by the European Union's Seventh Framework Programme for research, technological development and demonstration under Marie Curie grant agreement no 608695. MS and BHS were funded through the Polish-Norwegian Research Programme project iAREA, and RCN projects AEROCOM-P3 and AC/BC (240372). RCE and SJG were supported by the U. S. Department of Energy Office of Science Decadal and

Regional Climate Prediction using Earth System Models (EaSM) program. The Pacific Northwest National Laboratory is operated for the DOE by Battelle Memorial Institute under contract DE-AC06-76RLO 1830. GL and JEP were supported by the National Science Foundation (projects AGS-0946739 and ARC-1023387) and the Department of Energy (DOE FG02 01 ER63248 and DOE DE-SC0008486). TT is supported by the NEC SX-ACE supercomputer system of the National Institute for Environmental Studies, Japan, the Environmental Research and Technology Development Fund (S-12-3) of the Ministry of

Environment, Japan and JSPS KAKENHI Grant Numbers JP15H01728 and JP15K12190. TI, AK and ØS were supported by the Norwegian Research Council through the projects EVA (grant 229771), EarthClim (207711/E10), NOTUR (nn2345k), and NorStore (ns2345k), and through the EU projects PEGASOS and ACCESS and Nordforsk-CRAICC. P.S. acknowledges funding from the European Research Council (ERC) under the European Union's Seventh Framework Programme (FP7/2007–2013) ERC project ACCLAIM (Grant Agreement FP7-280025) and from the UK Natural Environment Research Council

project GASSP (grant number NE/J022624/1). The CESM project is supported by the National Science Foundation and the Office of Science (BER) of the U.S. Department of Energy. NCAR is sponsored by the National Science Foundation. CALIPSO is a joint satellite mission between NASA and the French Agency, CNES. We thank the PI investigators and their staff for establishing and maintaining the AERONET sites used in this study.

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





**Table 1: List of the models used in this study and which species they have reported.**

| | Aerosol Optical Depth | | | | | | | Direct Aaerosol Effect | | | | | | |
| | Total | Sulfate | BC | OA | SOA | Nitrate | Dust | Sea-salt | Total | Sulfate | BC FF | OA FF | BB | SOA | Nitrate |
|---|---|---|---|---|---|---|---|---|---|---|---|---|---|---|---|
| CAM4-Oslo | x | x | x | x | | | x | x | x | | | | | | |
| HadGEM2 | x | x | x | x | x | x | x | x | x | x | x | x | x | | x |
| ECHAM5-HAM | x | x | x | | x | | x | x | x | x | x | x | x | x | |
| OsloCTM2 | x | x | x | x | x | x | x | x | x | x | x | x | x | x | x |
| SPRINTARS | x | x | x | x | x | | x | x | x | x | x | x | x | | |
| GISS-MATRIX | x | | | | | | | | x | x | x | x | | | |
| GISS-modelE | x | x | x | x | x | | x | x | x | x | x | x | x | | |
| CAM5.1 | x | x | x | x | x | | x | x | x | x | x | x | x | x | |
| BCC | x | x | x | x | | | | | x | x | x | x | x | | |
| GMI-MERRA-v3 | x | x | x | x | | x | | x | x | x | x | x | x | | x |
| GEOS-Chem | x | | | | | | x | x | x | x | x | x | x | x | x |
| GOCART-v4 | x | x | x | x | | | x | x | x | x | x | x | x | | |
| NCAR-CAM3.5 | x | | | | | | | | x | x | x | x | x | | x |
| IMPACT | x | | | | | | | | x | x | x | x | x | x | |
| INCA | x | x | x | x | | | x | x | x | x | x | x | x | | x |
| TM5-V3 | x | x | x | x | | x | x | x | x | | | | | | |

**Table 2: List of the Arctic and Antarctic stations with ground-based measurements of AOD. Data for Tiksi, Andenes, Yakutsk, Bonanza Creek, Resolute Bay, and Kangerlussuaq are taken from the AERONET data base (http://aeronet.gsfc.nasa.gov/) and data from Ny Ålesund, Barrow and Alert are from Stone et al. (2014) and Tomasi et al. (2014).**

| Stations | Coordinates and altitude (amsl) | Measurement period |
|---|---|---|
| Tiksi | N 71°, E 128°, Alt 0 m | 2010-2012, 2014 |
| Andenes | N 69°, E 16°, Alt 379 m | 2002, 2008-2011, 2013, 2014 |
| Yakutsk | N 61°, E 129°, Alt 118 m | 2004-2015 |
| Bonanza Creek | N 64°, W 148°, Alt 150 m | 1994- 1997, 1999-2015 |
| Resolute_Bay | N 74°, W 94°, Alt 40 m | 2004, 2006, 2008-2015 |
| Kangerlussuaq | N 66°, W 50°, Alt 320 m | 2008-2015 |
| Ny Ålesund | N 78°, E 11°, Alt 5 m | 2001-2011 |



| Barrow | N 71°, W 156°, Alt 8 m | 2001-2011 |
| Alert | N 82°, W 62°, Alt 210 m | 2004-2011 |
| Neumeyer | S 70°, W 8°, Alt 40 m | 2000-2007 |
| Troll | S 72°, E 2°, Alt 1309 m | 2007-2013 |
| South Pole | S 90°, E 0°, Alt 2835 m | 2001-2012 |

**Table 3: Annual mean Arctic (60° N - 90° N) AOD AeroCom Phase II model-median, model range (minimum and maximum), and the 25th/75th percentile. The number of models for each specie is given in the rightmost column. BC is total BC from all sources and OA is total OA from all sources.**

| | Median | Minimum | Maximum | 25th percentile | 75th percentile | Number of models |
|---|---|---|---|---|---|---|
| Total | 0.071 | 0.025 | 0.183 | 0.031 | 0.121 | 16 |
| Sulfate | 0.032 | 0.003 | 0.068 | 0.009 | 0.044 | 12 |
| BC | 0.002 | 0.000 | 0.004 | 0.001 | 0.003 | 12 |
| OA | 0.014 | 0.001 | 0.072 | 0.006 | 0.017 | 11 |
| SOA | 0.004 | 0.001 | 0.012 | 0.002 | 0.007 | 6 |
| Nitrate | 0.001 | 0.000 | 0.011 | 0.000 | 0.003 | 4 |
| Sea-salt | 0.013 | 0.001 | 0.054 | 0.007 | 0.018 | 12 |
| Dust | 0.008 | 0.001 | 0.035 | 0.003 | 0.014 | 11 |

**Table 4: As Table 3, but averaged over the Antarctic region (70° S - 90° S).**

| | Median | Minimum | Maximum | 25th percentile | 75th percentile | Number of models |
|---|---|---|---|---|---|---|
| Total | 0.014 | 0.001 | 0.052 | 0.003 | 0.019 | 16 |
| Sulfate | 0.007 | 0.000 | 0.014 | 0.000 | 0.009 | 12 |
| BC | 0.000 | 0.000 | 0.001 | 0.000 | 0.000 | 12 |
| OA | 0.001 | 0.000 | 0.020 | 0.000 | 0.002 | 11 |
| SOA | 0.001 | 0.000 | 0.003 | 0.000 | 0.001 | 6 |
| Nitrate | 0.000 | 0.000 | 0.002 | 0.000 | 0.000 | 4 |
| Sea-salt | 0.002 | 0.000 | 0.032 | 0.001 | 0.011 | 12 |
| Dust | 0.001 | 0.000 | 0.002 | 0.000 | 0.002 | 11 |

**Table 5: Annual mean Arctic (60° N - 90° N) DAE AeroCom Phase II model-median, model range (minimum and maximum), and the 25th/75th percentile. The number of models for each specie is given in the rightmost column. BC FF is BC from fossil fuel and biofuel emissions and OA FF is OA from fossil fuel and biofuel emissions. BB is BC and AO combined from biomass burning emissions.**




|  | Median | Minimum | Maximum | 25th percentile | 75th percentile | Number of models |
|---|---|---|---|---|---|---|
| Total | -0.12 | -0.30 | 0.09 | -0.22 | 0.01 | 16 |
| Sulfate | -0.24 | -0.43 | 0.01 | -0.29 | -0.11 | 14 |
| BC FF | 0.19 | 0.03 | 0.37 | 0.12 | 0.26 | 14 |
| OA FF | 0.00 | -0.04 | 0.02 | -0.02 | 0.00 | 14 |
| BB | 0.01 | -0.06 | 0.04 | -0.02 | 0.02 | 13 |
| SOA | -0.01 | -0.12 | 0.01 | -0.02 | 0.00 | 5 |
| Nitrate | -0.03 | -0.09 | 0.00 | -0.06 | -0.01 | 6 |

**Table 6: As Table 5, but averaged over the Antarctic region (70° S - 90° S).**

|  | Median | Minimum | Maximum | 25th percentile | 75th percentile | Number of models |
|---|---|---|---|---|---|---|
| Total | 0.03 | 0.00 | 0.10 | 0.01 | 0.07 | 16 |
| Sulfate | 0.00 | -0.03 | 0.00 | -0.01 | 0.00 | 14 |
| BC FF | 0.02 | 0.00 | 0.09 | 0.01 | 0.04 | 14 |
| OA FF | 0.00 | 0.00 | 0.01 | 0.00 | 0.00 | 14 |
| BB | 0.01 | -0.01 | 0.08 | 0.00 | 0.02 | 13 |
| SOA | 0.00 | 0.00 | 0.01 | 0.00 | 0.00 | 5 |
| Nitrate | 0.00 | 0.00 | 0.03 | 0.00 | 0.00 | 6 |

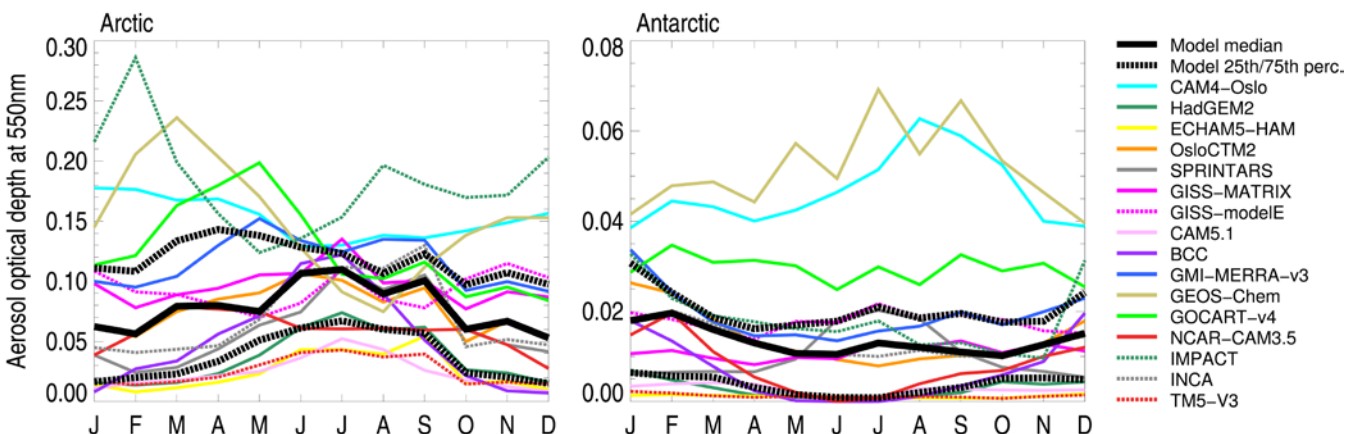

**Figure 1: Mean seasonal cycle of the Arctic (60° N - 90° N) (left) and Antarctic (70° S - 90° S) right) total AOD. The different colours represent the different AeroCom Phase II models. The black solid line is the model-median and the dashed line is the 25th/75th percentile.**





**Figure 2: Seasonal cycle of model-median AOD compared to observations for nine Arctic stations; (a) Alert (82° N, 62° W), (b) Ny-Ålesund (78° N, 11° E), (c) Barrow (71° N, 156° W) (d) Kangerlussuaq (66° N, 50° W) (e) Resolute Bay (74° N, 94° W) (f) Bonanza Creek (64° N, 148° W), (g) Yakutsk (61° N, 129° E), (h) Andenes (69° N, 16° E), and (i) Tiksi (71° N, 128° E). The black solid line is the model-median and the black dashed line is the 25th/75th percentile. Models are shown in thin, grey lines. The red solid line is the observational mean and the dashed red line is one standard deviation from mean values. Measurements for (a) – (c) are taken Stone et al. (2014), (d)-(i) are from AERONET stations.**





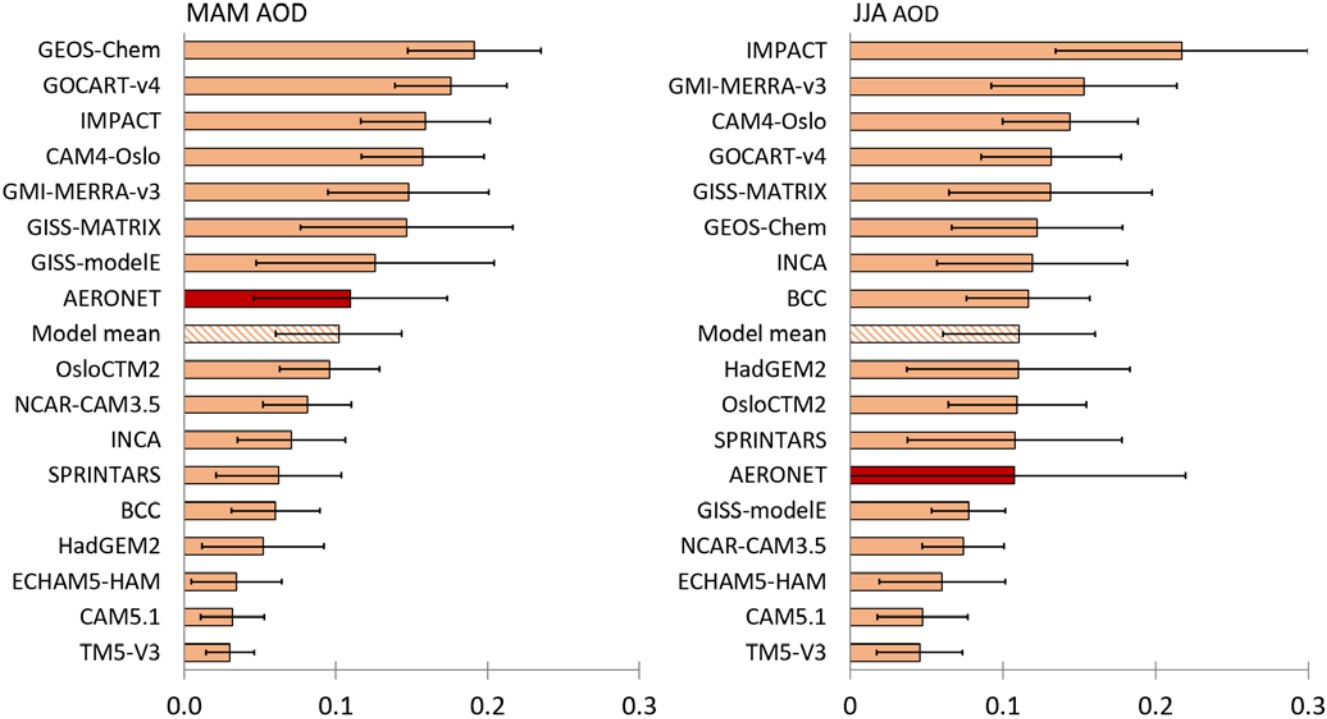

**Figure 3: Multi-model AOD averaged for the spring (MAM) and summer (JJA) for the 9 Arctic AERONET stations in Fig. 2 (details Table 2). The red bar is the average over the 9 stations and the striped bar is the multi-model average. The error bars represent one standard deviation.**

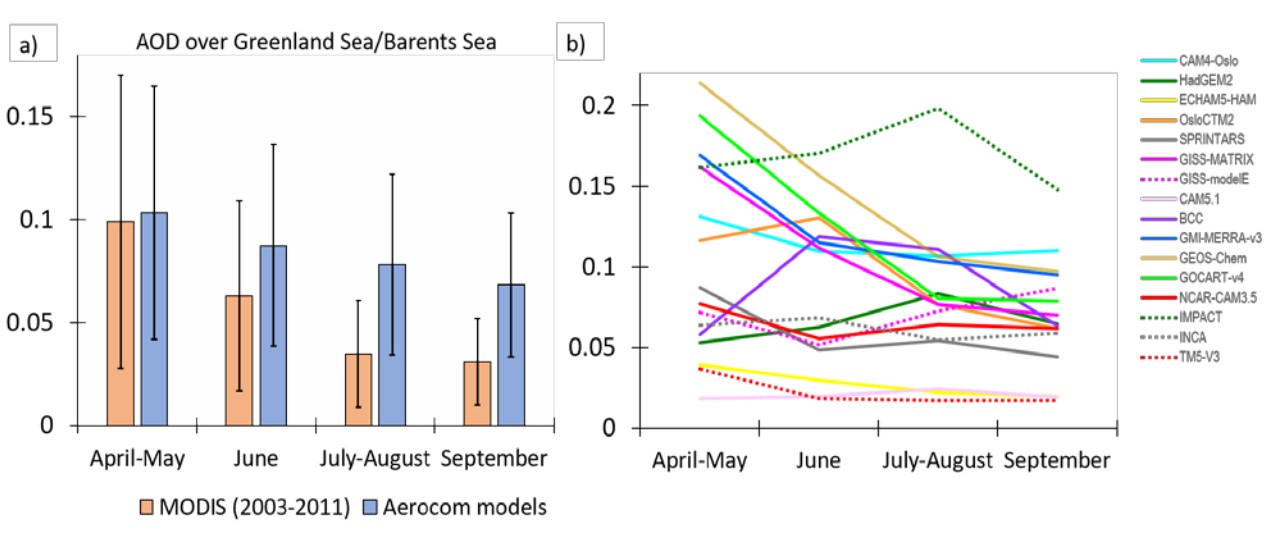



**Figure 4: AOD over the Arctic Ocean (75° N - 82° N, 10° W - 40° E) from a) MODIS retrieval 2003-2011 median (orange) compared to the AeroCom phase II model mean (blue) and b) the individual models. The error bars in a) represent one standard deviation. Note the different axes.**

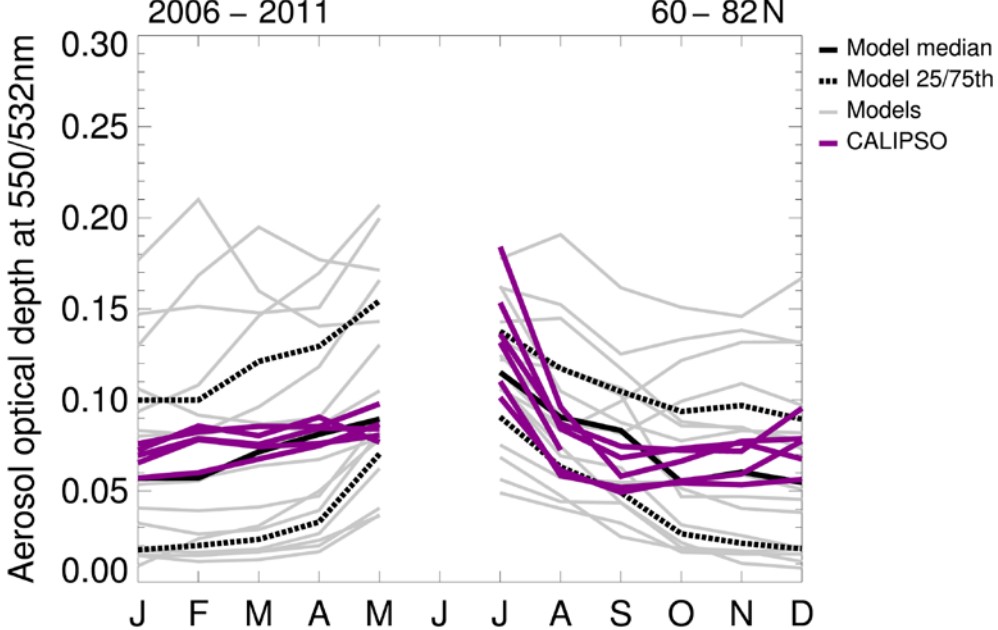

5   **Figure 5: AOD in the Arctic (60° N - 82° N) from CALIPSO retrieval for the different years 2006-2011 (AOD532nm) (purple lines) compared to the AeroCom phase II models (thin grey lines) screened by CALIOP availability.**

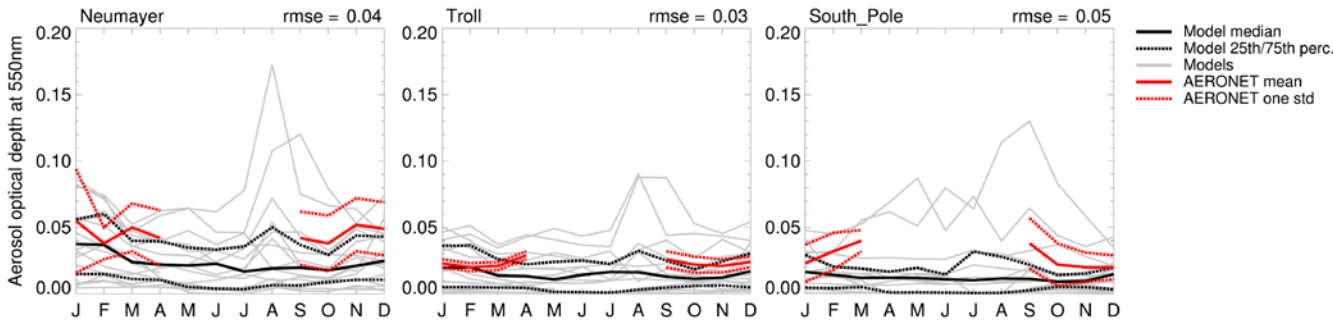

**Figure 6: As Fig. 2, but for three Antarctic sites: Neumeyer 70° S, 8° W (left), Troll 72° S, 2° E (middle), and South Pole 90° S, 0° E**
10  **(right), with values from Tomasi et al. (2015).**



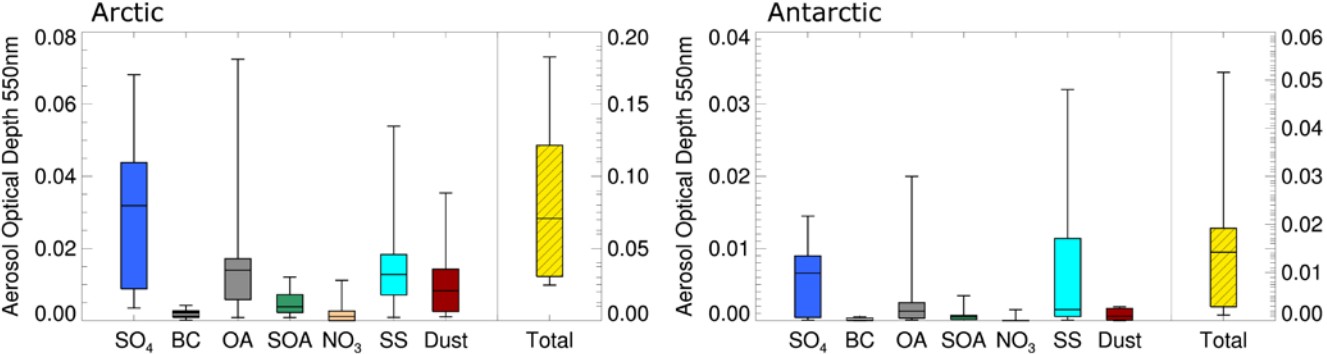

**Figure 7: Annual mean multi-model median AOD for the individual components (sulfate, BC (from all sources), total OA (from all sources), SOA, nitrate, sea-salt, and dust) and the total aerosol, averaged over the Arctic (60° N - 90° N) (left) and the Antarctic (70° S - 90° S) (right). The bottom and top of each box are the first and third (25th/75th) quartiles, and the band inside the box is the model-median. The whiskers represent the minimum and maximum of the model range. Note the different vertical axis; the right-side vertical axis is for total AOD.**

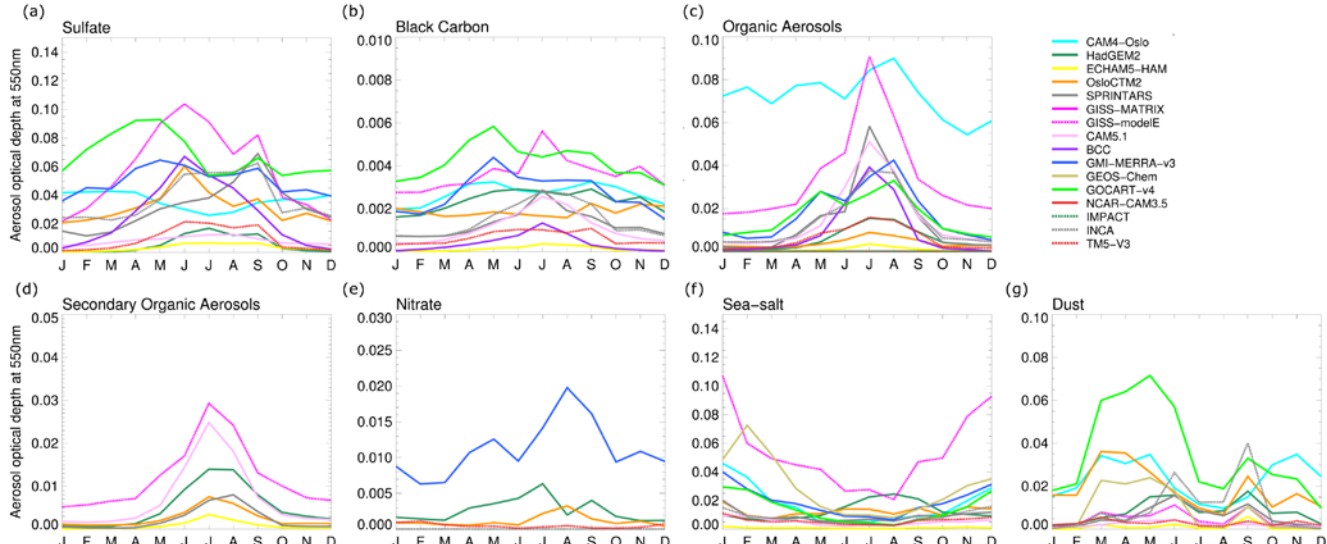

**Figure 8: Seasonal cycle multi-model Arctic AOD for (a) sulfate, (b) BC (from all sources), (c) total OA (from all sources), (d) SOA, (e) nitrate, (f) sea-salt, and (g) dust. The Arctic region is defined as 60° N - 90° N. Note the different axes.**





**Figure 9: The geographical distributed model-median Arctic AOD (a) sulfate, (b) BC (from all sources), (c) total OA (from all sources), (d) SOA, (e) nitrate, (f) sea-salt, and (g) dust, for the summer (JJA) season (left) and the winter (DJF) season (right). Note the different axes.**





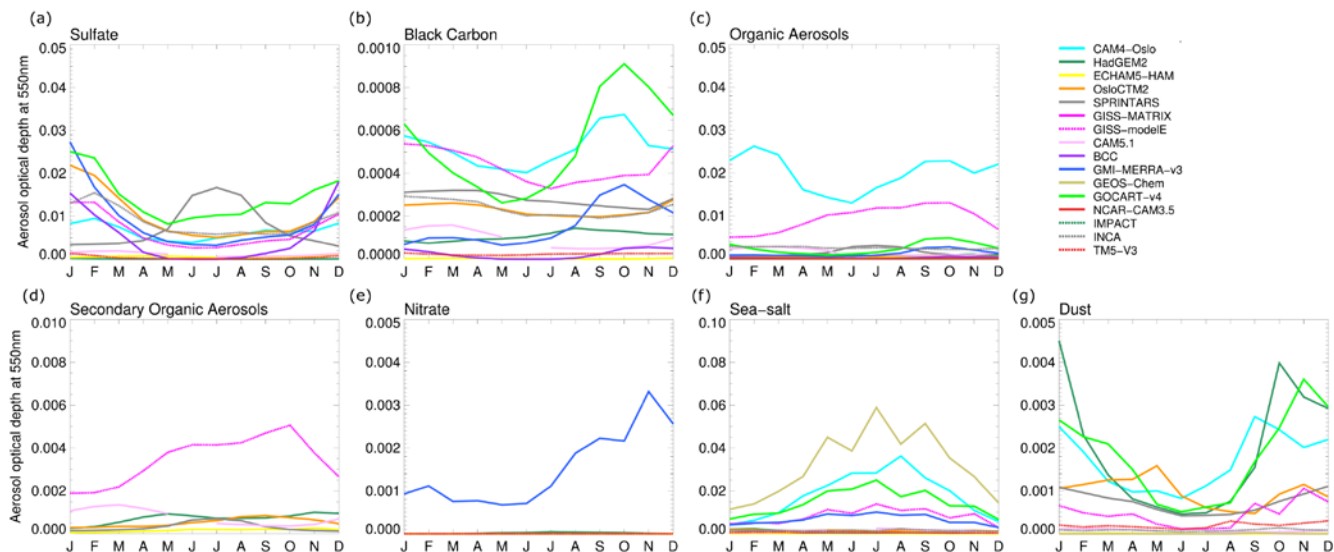

**Figure 10: Multi-model seasonal cycle Antarctic mean (70° S - 90° S) AOD for the individual components (a) sulfate, (b) BC (from all sources), (c) total OA (from all sources), (d) SOA, (e) nitrate, (f) sea-salt and (g) dust. Note the different axes.**

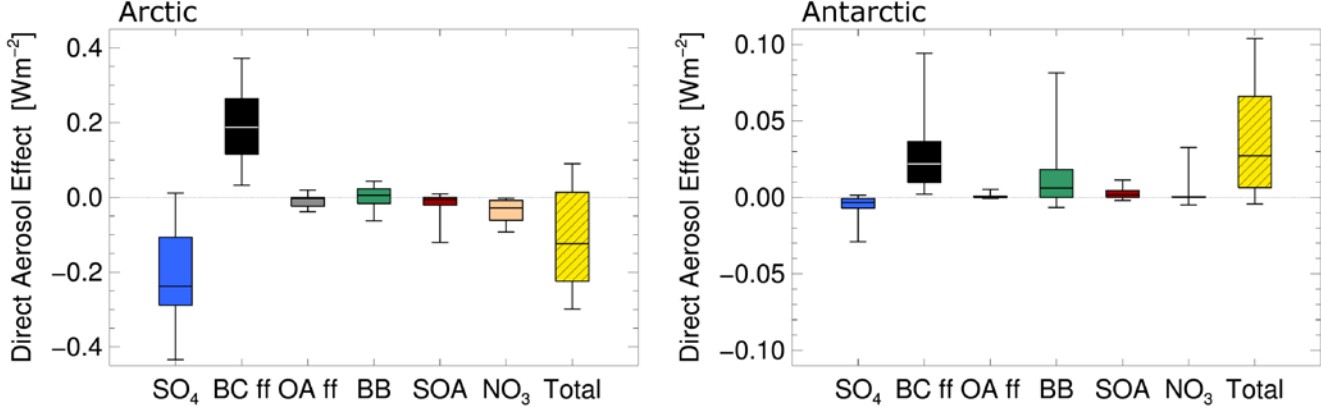

5    **Figure 11: Annual mean multi-model DAE (in W m⁻²) averaged over the Arctic (60° N - 90° N) (left) and the Antarctic (70° S - 90° S) (right), for the individual components (sulfate, BC FF, OA FF, BB, SOA, and nitrate) and the total aerosol. The bottom and top of each box are the first and third (25th/75th) quartiles, and the band inside the box is the model-median. The whiskers represent the minimum and maximum of the model range. Note the different vertical axis.**





**Figure 12: Arctic mean DAE seasonal cycle and the summer (JJA) mean model-median geographical distribution of (a) total aerosol, (b) sulfate, (c) BC FF, (d) OA FF, (e) BB, (f) SOA, and (g) nitrate. Note the different axes.**



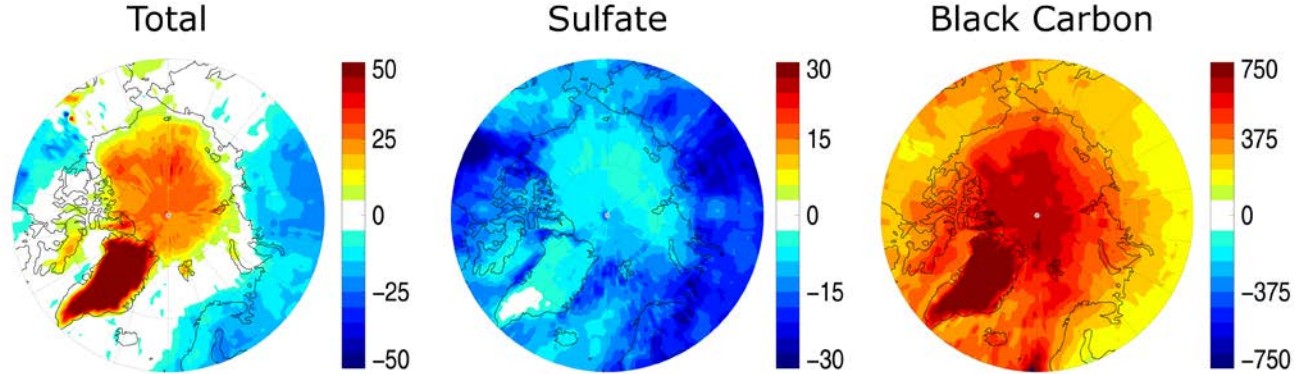

**Figure 13: JJA mean model-median normalized DAE, i.e. total DAE (in W m⁻²) per total AOD (left), sulfate DAE per sulfate AOD (middle) and BC FF DAE per BC AOD (right). Note the different axes.**

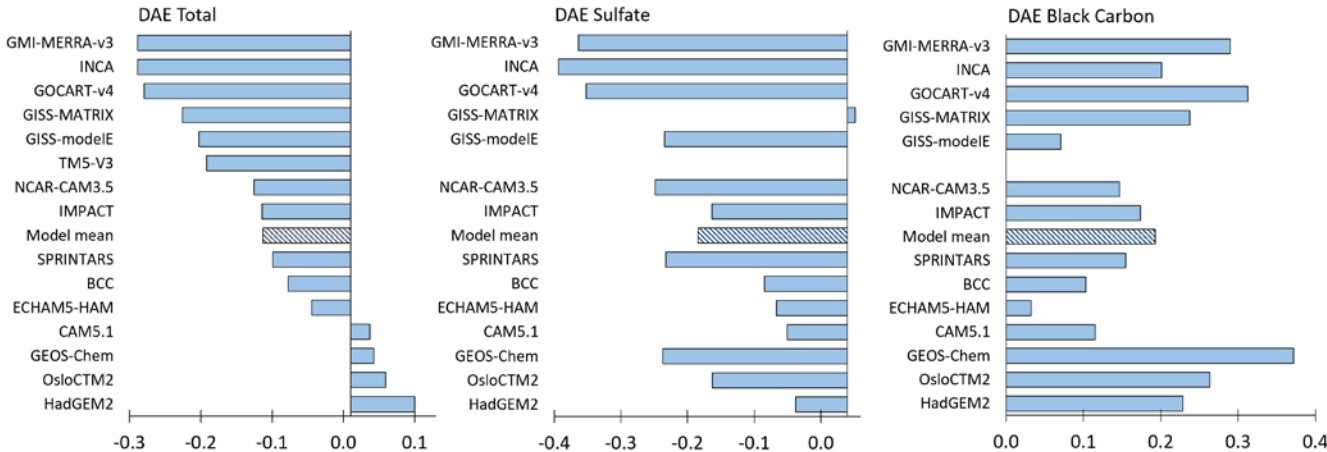

5    **Figure 14: Arctic annual mean DAE (in W m⁻²) for the AeroCom phase II models, TOTAL (left), sulfate (middle), BC FF (right). The striped bar is the model mean. TM5-V3 have reported total only.**



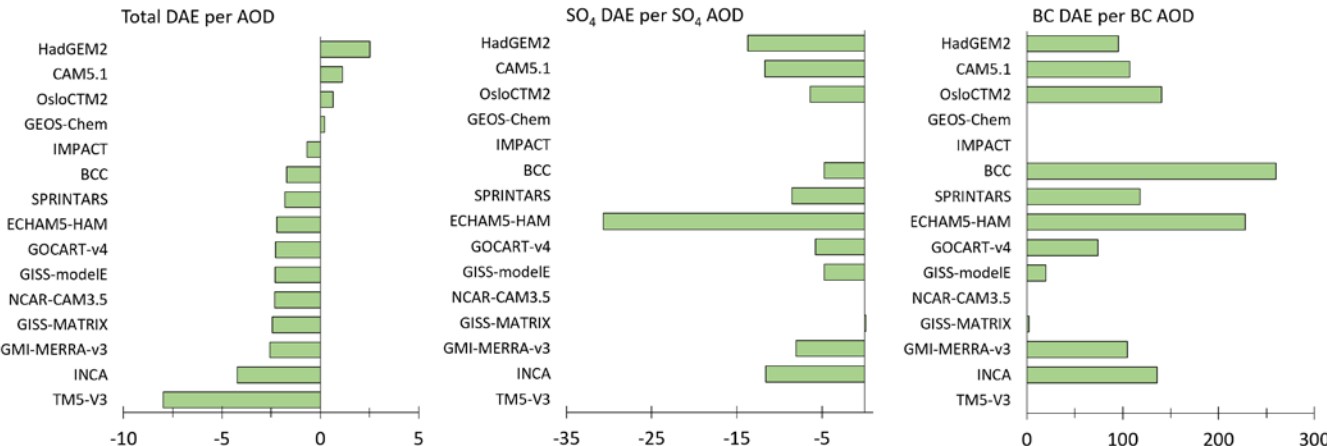

**Figure 15: Arctic annual mean DAE per AOD for all AeroCom phase II models, TOTAL (left), sulfate (middle), and BC FF (right). Values for sulfate and BC is missing for GEOS-Chem, IMPACT, NCAR-CAM3.5, and TM5 (see Table 1).**

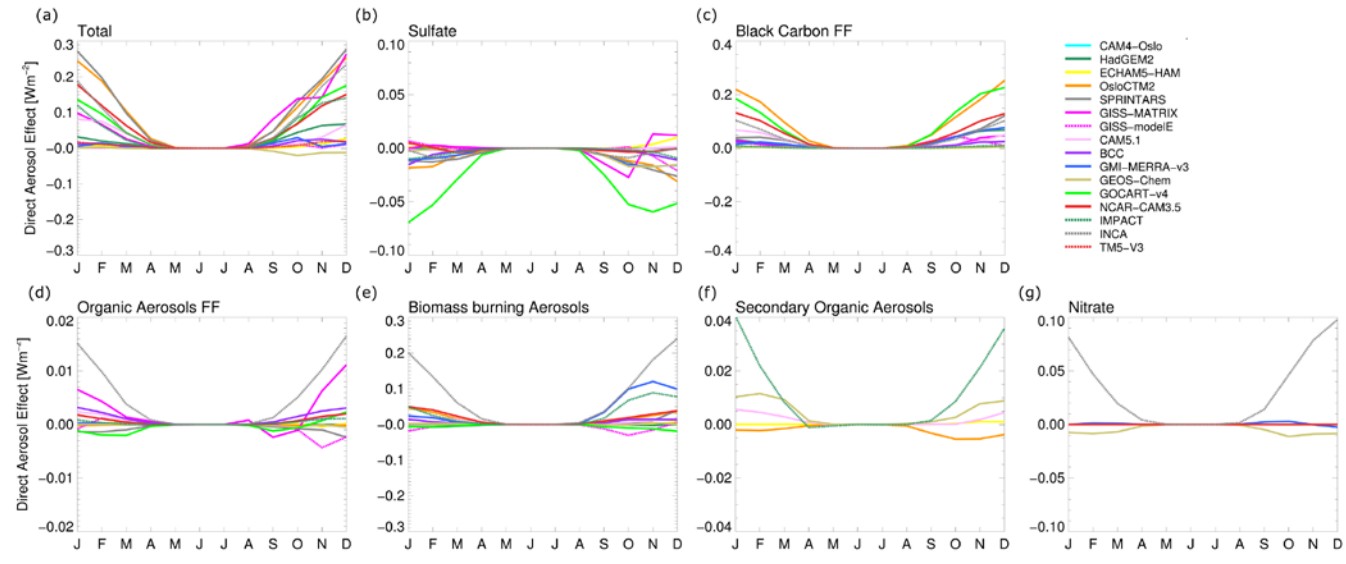

**Figure 16: Antarctic mean (70° S - 90° S) seasonal cycle of (a) the total DAE and the DAE for (b) sulfate, (c) BC FF, (d) OA FF, (e) BB, (f) SOA, and (g) nitrate. Note the different axes.**





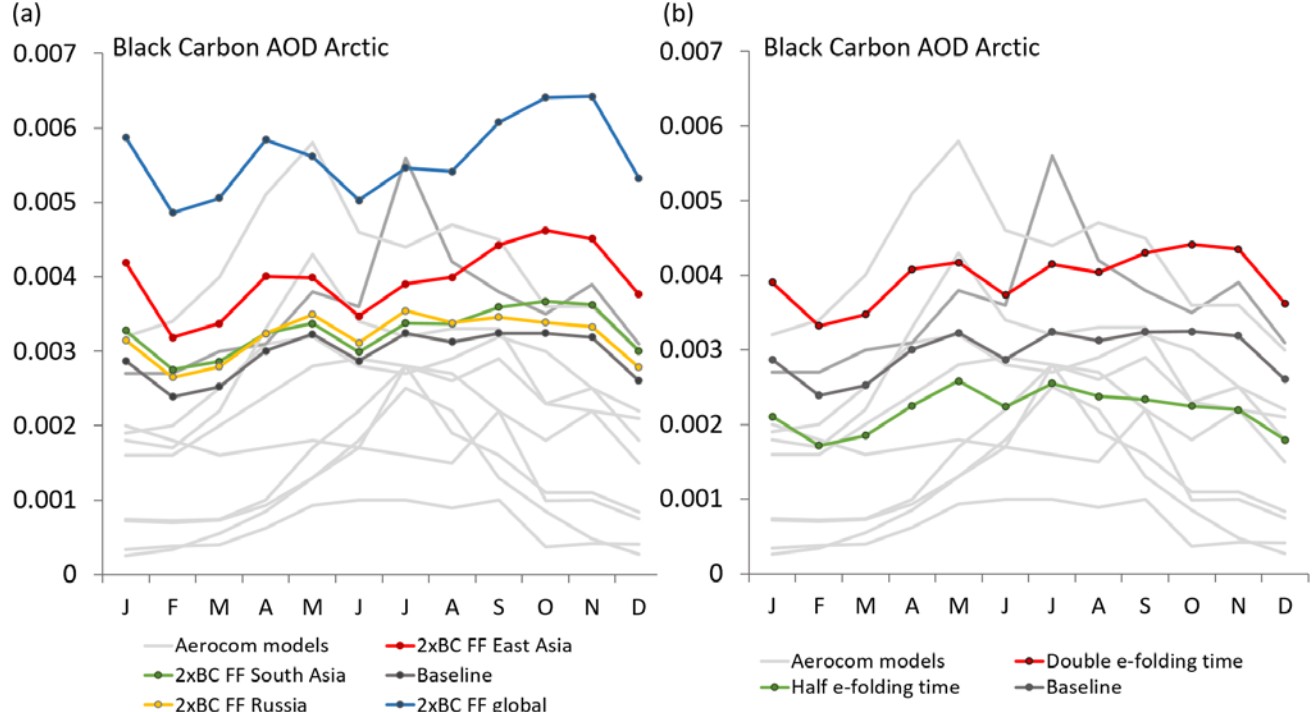

**Figure 17: Arctic mean seasonal cycle of (total) BC AOD for simulations with GISS modelE (in colors) compared to the AeroCom models (in light grey). The darker grey AeroCom model is the GISS modelE AeroCom run. (a) shows emission perturbations for a doubling of BC emissions (fossil fuel and biofuel) in South Asia (green), East Asia (red), Russia (yellow), and global (blue). (b) shows double (red) and half (green) of the e-folding time from hydrophobic to hydrophilic BC.**