# Peer review of "Aerosols at the Poles: An AeroCom Phase II multi-model evaluation"

_Atmospheric Chemistry and Physics, 2016_

## Referee Comment (RC1) · Anonymous Referee #1 · 30 Mar 2017

The paper describe a important topic related to the behaviour of aerosol modelling at Poles. A comparison between models and most important with observations is reported. This add value to the manuscript. However, prior the publication a deeper description related not only to the observed behaviour of the models, but also to the underlying mechanisms (i.e. reasons) is required throughout the whole paper. Here below some suggestion to improve the paper.

MAJOR POINTS

Page 5, lines 4-5. Even the "Model descriptions including model resolution, dynamics, and microphysics schemes used are given in Table 1 and 2 in Myhre et al. (2013)", it should be useful for the reader to have at disposal the main model features in the present paper. I suggest to resume them in the supplemental material

[Figure]

Page 5, lines 6-7. "Each model has provided climate and aerosol simulations using year 2006 meteorology. For present-day simulations emissions for year 2000 have been used, and for preindustrial runs year 1850 emissions have been used (Lamarque et al., 2010)." If the assumption on the used datasets for emissions appears reliable, it is not so intuitive for the reader to understand the choice of the year 2006 as a reference for meteorology. Please, demonstrate that 2006 year does not present any anomaly for what concern meteorology and general atmospheric circulation compared at least to the 2000-2015 period (the period of experimental data used for the comparison) using both modelling and experimental data i.e. GAW-WMO data. It seems that the 2006 assumption represents a big limitation when compared the simulated AOD with AERONET data (Figure 2). A big effort has to be done to overcome this limitation or to details its implication in term of uncertainty on the simulated aerosol properties when compared to other dataset based on multi-annual data. For example Figure 5 seems to overcome this limitation due to the fact that only 2006-2007 CALIPSO data were used.

Page 5, lines 17-18: "The models have estimated AOD as a combination of aerosol abundancies and optical properties, which is why AOD can be reported in the months where there is no actual sunlight". Please details (at least in supplemental material) how each model calculates the aerosol optical properties and the underlying assumptions (i.e. mixing state, hygroscopic growth etc.).

Page 7, line 28 – page 8, lines 1-3: "For GEOS-Chem and GOCART this maximum is dominated by natural aerosols (sea-salt and dust, respectively, as shown in Fig. 8). Note that modeled AOD is calculated from simulated aerosol distributions, and can therefore be reported even for months where there is no actual sunlight." This sentence is limited just to observe that the behaviour of GEOS-Chem and GOCART is related to wrong simulation of sea-salt and dust. But it should be important to describe the inner reason for the overestimation of sea-salt and dust. Could you describe them?

Figure 1b: which is the reason for the overestimation of CAM4-Oslo in the Antarctic? Please add to the text an explanation.

Figure 5 is very promising. May I suggest also to add a comparison concerning the vertical behaviour of aerosol extinction coefficient? As the simulated radiative forcing strongly depends from the aerosol vertical profiles, this comparison could be very useful.

Figure 10 and related description at page 10 lines 5-7: the description is too short to capture the complexity of Figure 10. Please improve it. For example: why GISS-MATRIX forecast a such big contribution of SOA compared to other models? Why GMI-MERRA-v3 did the same for Nitrate? What about the SPRINTARS behaviour for sulphate which is opposed to the other models? A deeper description related not only to the observed behaviour but also the underlying mechanisms (i.e. reasons) is required here and throughout the whole paper.

Section 3.2: is the calculated radiative forcing in clear sky approximation or in all sky conditions?

MINOR POINT

Page 10, line 13: "although the Arctic AOD of BC sis". Change "sis" with "is".

---

## Referee Comment (RC2) · Anonymous Referee #2 · 21 Apr 2017

This is a comprehensive study comparing a multi-model ensemble of AOD with ground based and satellite observation. Analysis are performed for both poles and for different aerosol components. The analysis is performed in good detail and shows that there is a wide model spread and the model median predicts the observation well. I recommend this paper for publication after some modifications:

A map showing where all the locations of the stations are would be useful. In the map also the area over which the satellite comparison has been performed could be added.

In the introduction you write: Arctic AOD typically has a maximum during late winter and spring. In the data you show this is only true for some stations (e.g. Fig 2) and only clear for dust and sea-salt (Fig 8). Why can't this typical maximum not be seen in black carbon or sulfate? In Figure 2 only some stations show the spring peak - is

this based on the location of the station, or is it a specific component which causes it and is not transported to the other stations. Would you expect, that the spring peak is stronger for surface or for the total column? I suggest to add some further description focusing on the seasonal cycle as well as the horizontal distribution of the aerosol.

For the sensitivity study you change the emission and lifetime and look at the impact on AOD in the Arctic. To put this better into context: How long is the average lifetime in all the models compared to the case study model and how does the lifetime change over the seasons? Do you expect the BC emissions used are too low, or the lifetime is too short, can the sensitivity study help answering this?

Comparing Fig 4 and 5 it seems that MODIS retrieved AOD are twice as high in April compared to August. In the CALIPSO AOD July values are double as high as April value. Is this big differences based to the screening in MODIS, you mention? If you could add a panel like Fig 4a) to Fig 5, it might be easier to see how well the model capture the different annual cycle.

here are some further comments, given for the page and line number:

p1, li30 - there is a too abrupt change from transport to radiative forcing to AOD, you could add a sentence connecting those two topics.

p2, li 1 - in winter

p2, li 11 - add also the percentage of the changes due to doubling the lifetime

p2, li 23 - it should be "Faluvegi", you wrote "Feluvegi"

p3, li 28 - Can you give a reference for aerosols transported into the Antarctic?

p5, li 7 - for "the" year 2000

p5, li 9 - what do you mean with "double calls"? please explain in more detail or rephrase.

p5, li 10 - species is the singular form of species, the word specie appears also later in the text, please change.

p5, li 12 - biomass "burning" emissions (correct?) - this also appears later in the text

p7, li 25 - of "the" year 2000

p7, li 25 - It would be easier to understand if you also refer to the figure (Fig 1, black thick line)

p8, li 10 - r could be reported in the figure next to the rmse, so it is available for each stations

p8, li 17 - either write "high reflectivity" or "highly reflective"

p8, li 19 - (see methods section)

p8, li 23 - you give the uncertainty range here, how is it calculated?

p8, li 25 - Some of the models "do" have a steeper slope

p9, li 7 - You explain how the rms was calculated, the rmse value should also appear the text when discussing them.

p 12, li 7 - it is written that you don't include semi- or indirect cloud effects, or surface albedo modifications, how much approximately would the cloud effects and surface albedo modification impact the DAE?

p 12, li 17 - you give a percentage of the effect by doubling emission, what would be the corresponding value for the changes in lifetime? (also in the abstract)

Figures:

Fig 3, caption: the average "observation" over the 9 stations

Fig 14, caption: "total" in small case

Fig 8: also here the multi model median could be added.

Fig 17: you could add the information that the left panel shows the experiment with doubling emission, while the right panel shows the effect of the changes in e-folding lifetime in the figure title.

---

## Author Response (AR1)

We would like to thank the reviewer for the comments and feedbacks to help improve this manuscript!

**Anonymous Referee #1**

*The paper describe a important topic related to the behaviour of aerosol modelling at Poles. A comparison between models and most important with observations is reported. This add value to the manuscript. However, prior the publication a deeper description related not only to the observed behaviour of the models, but also to the underlying mechanisms (i.e. reasons) is required throughout the whole paper. Here below some suggestion to improve the paper.*

*MAJOR POINTS*

*Page 5, lines 4-5. Even the "Model descriptions including model resolution, dynamics, and microphysics schemes used are given in Table 1 and 2 in Myhre et al. (2013)", it should be useful for the reader to have at disposal the main model features in the present paper. I suggest to resume them in the supplemental material*

We have made a new table describing the model resolution, meteorology, and aerosol microphysics (mixing state, size distribution and growth factors). The table is in the Supplementary (Table S1).

*Page 5, lines 6-7. "Each model has provided climate and aerosol simulations using year 2006 meteorology. For present-day simulations emissions for year 2000 have been used, and for preindustrial runs year 1850 emissions have been used (Lamarque et al., 2010)." If the assumption on the used datasets for emissions appears reliable, it is not so intuitive for the reader to understand the choice of the year 2006 as a reference for meteorology. Please, demonstrate that 2006 year does not present any anomaly for what concern meteorology and general atmospheric circulation compared at least to the 2000-2015 period (the period of experimental data used for the comparison) using both modelling and experimental data i.e. GAW-WMO data. It seems that the 2006 assumption represents a big limitation when compared the simulated AOD with AERONET data (Figure 2). A big effort has to be done to overcome this limitation or to details its implication in term of uncertainty on the simulated aerosol properties when compared to other dataset based on multi-annual data. For example Figure 5 seems to overcome this limitation due to the fact that only 2006-2007 CALIPSO data were used.*

The AeroCom model intercomparison setup is designed to have all the models run the same year. In that way, the transport will not be 'smoothed out' as in a climatology, and we can easier see if the models have the same features. Of course, you will then depend on the chosen year for transport. The AERONET stations does not provide enough data for 2006 to do a meaningful comparison, so therefore we included the climatology instead.

However, we have added data from 2006 from the stations where it was available, and we have plotted them together with the climatology in Figure 2.

*Page 5, lines 17-18: "The models have estimated AOD as a combination of aerosol abundancies and optical properties, which is why AOD can be reported in the months where there is no actual sunlight". Please details (at least in supplemental material) how each model calculates the aerosol optical properties and the underlying assumptions (i.e. mixing state, hygroscopic growth etc.).*

We have added this information in the new table in Supplementary.

*Page 7, line 28 – page 8, lines 1-3: "For GEOS-Chem and GOCART this maximum is dominated by natural aerosols (sea-salt and dust, respectively, as shown in Fig. 8). Note that modeled AOD is calculated from simulated aerosol distributions, and can therefore be reported even for months where there is no actual sunlight." This sentence is limited just to observe that the behaviour of GEOS-Chem and GOCART is related to wrong simulation of sea-salt and dust. But it should be important to describe the inner reason for the overestimation of sea-salt and dust. Could you describe them?*

We do not know if these models have 'wrong simulation' of sea-salt and dust. These models might as well be closer to the real atmosphere than the other models. Unfortunately, we have no observations of dust AOD we can compare with; we only have observations for total AOD.

*Figure 1b: which is the reason for the overestimation of CAM4-Oslo in the Antarctic? Please add to the text an explanation.*

We have added the following: 'The higher values of AOD in CAM-Oslo are linked to efficient vertical transport in deep convective clouds which exaggerates the amount of aerosols in the upper troposphere (and poleward transported aerosols)'

*Figure 5 is very promising. May I suggest also to add a comparison concerning the vertical behaviour of aerosol extinction coefficient? As the simulated radiative forcing strongly depends from the aerosol vertical profiles, this comparison could be very useful.*

We agree that it would be interesting to investigate the vertical profiles. However, there is only a subset of models that have provided vertical profiles of the extinction coefficient and Koffi et al. (2016) have already studies these in detail, although not with a polar focus.

We have added the following: 'Uncertainties in calculating the radiative impact of aerosols are linked to the vertical distribution of aerosols (Samset et al., 2013; Kipling et al., 2015). A comparison of the aerosol vertical extinction coefficient from 11 AeroCom models to CALIPSO has been performed in Koffi et al., (2016), showing large spread among the models.'

*Figure 10 and related description at page 10 lines 5-7: the description is too short to capture the complexity of Figure 10. Please improve it. For example: why GISSMATRIX forecast a such big contribution of SOA compared to other models? Why GMI-MERRA-v3 did the same for Nitrate? What about the SPRINTARS behavior for sulphate which is opposed to the other models? A deeper description related not only to the observed behaviour but also the underlying mechanisms (i.e. reasons) is required here and throughout the whole paper.*

A multi model comparison study like this cannot go deep into the reasons and underlying mechanisms behind the overestimation or underestimation of different species. For that (single) model studies which can perform dedicated sensitivity tests to isolate causes are needed. We have tried to include more information about why some of the models stand out, but the underlying mechanisms can be multiple:

'GISS-modelE shows higher values for SOA. This model has the highest SOA lifetime (14 days) because of large amounts of SOA in the upper troposphere where there is less scavenging and more SOA available to be transported poleward.'

'SPRINTARS shows opposite behavior on the seasonal cycle of sulfate AOD compared to the other models. This is likely linked to an anomaly in the relative humidity over East Antarctica in the simulation and has been improved in a newer version of the model.'

'GMI-MERRA shows higher nitrate AOD values compared to the other models and is probably linked to the inclusion of oceanic NH3 emissions (based on the GEIA emission inventory) in the model. The CMIP5 emission dataset do not include NH3 oceanic emissions.'

*Section 3.2: is the calculated radiative forcing in clear sky approximation or in all sky conditions?*

We have used the all-sky radiative forcing calculations. We have now specified this in the text.

*MINOR POINT*

*Page 10, line 13: "although the Arctic AOD of BC sis". Change "sis" with "is".*

Done.

We would like to thank the reviewer for the comments and feedback to help improve this manuscript!

**Anonymous Referee #2**

*This is a comprehensive study comparing a multi-model ensemble of AOD with ground based and satellite observation. Analysis are performed for both poles and for different aerosol components. The analysis is performed in good detail and shows that there is a wide model spread and the model median predicts the observation well. I recommend this paper for publication after some modifications:*

*A map showing where all the locations of the stations are would be useful. In the map also the area over which the satellite comparison has been performed could be added.*

This is a good idea; we have added a new figure showing the station locations and the satellite area (new Figure 3).

*In the introduction you write: Arctic AOD typically has a maximum during late winter and spring. In the data you show this is only true for some stations (e.g. Fig 2) and only clear for dust and sea-salt (Fig 8). Why can't this typical maximum not be seen in black carbon or sulfate? In Figure 2 only some stations show the spring peak – is this based on the location of the station, or is it a specific component which causes it and is not transported to the other stations. Would you expect, that the spring peak is stronger for surface or for the total column? I suggest to add some further description focusing on the seasonal cycle as well as the horizontal distribution of the aerosol.*

This is a good point. We would expect the spring peak to be stronger at the surface than for the total column, since the age of air, and the amplitude of its seasonal cycle decrease strongly with altitude (Stohl 2006). We have added some text explaining this.

Why models struggle to simulate the Arctic haze is a subject of many studies, and the reasons are multiple and model-dependent. In this model-intercomparison study we are not able to investigate why the models do not simulate a spring peak in BC and sulfate.

In the new figure of the locations of the stations (Fig. 3), we have added a pie chart for each station of the distribution of the different aerosols' AOD (MAM average). This is to better show the difference between the locations -as simulated in the models. We have rewritten the text in the Results chapter describing the AERONET observations of AOD.

*For the sensitivity study you change the emission and lifetime and look at the impact on AOD in the Arctic. To put this better into context: How long is the average lifetime in all the models compared to the case study model and how does the lifetime change over the seasons? Do you expect the BC emissions used are too low, or the lifetime is too short, can the sensitivity study help answering this?*

The global annual mean BC lifetime in the AeroCom models ranges from 3.8 days (CAM5.1) to 17.1 days (HadGEM2) (Samset et al., 2014). GISS modelE has a BC global lifetime of 5.9 days

close to the AeroCom average global lifetime of 6.5 days. We have added this to the text. We have not the lifetime for the Arctic, and we do not know how this lifetime changes over the seasons. As there are many models parameters that influence the Arctic BC burden, it is difficult to make any firm conclusions regarding the magnitude of emissions and/or lifetime based on these sensitivity tests. Also, tuning BC burden in the Arctic only, might lead to larger bias elsewhere.

*Comparing Fig 4 and 5 it seems that MODIS retrieved AOD are twice as high in April compared to August. In the CALIPSO AOD July values are double as high as April value. Is this big differences based to the screening in MODIS, you mention? If you could add a panel like Fig 4a) to Fig 5, it might be easier to see how well the model capture the different annual cycle.*

Yes, the reasons for the high values for CALIPSO in summer compared to MODIS is due to the screening of the data. In the summer months, only the southernmost latitude bands dominated by biomass burning are screened. We are not quite sure what the reviewer means by adding Fig4a to Fig5. We have tried different plotting options for Fig5, and we decided that this was the best way to show the seasonal variation of the models and CALIPSO.

*here are some further comments, given for the page and line number:*

*p1, li30 - there is a too abrupt change from transport to radiative forcing to AOD, you could add a sentence connecting those two topics.*

Done.

*p2, li 1 - in winter*

Done.

*p2, li 11 - add also the percentage of the changes due to doubling the lifetime*

We have added: 'A doubling of the BC lifetime, result in a 39 % increase in Arctic AOD of BC.'

*p2, li 23 - it should be "Faluvegi", you wrote "Feluvegi"*

Done.

*p3, li 28 - Can you give a reference for aerosols transported into the Antarctic?*

Done (Fiebig et al. (2009), Stohl and Sodemann (2010), and Tomasi (2015)).

*p5, li 7 - for "the" year 2000*

Done.

*p5, li 9 - what do you mean with "double calls"? please explain in more detail or rephrase.*

We have added the following explanation; ', i. e. for each time step the radiation code is called with and without the arguments needed to calculate the given species forcing,'

*p5, li 10 - species is the singular form of species, the word specie appears also later in the text, please change.*

Done.

*p5, li 12 - biomass "burning" emissions (correct?) - this also appears later in the text*

Yes, this has now been changed.

*p7, li 25 - of "the" year 2000*

Done.

*p7, li 25 - It would be easier to understand if you also refer to the figure (Fig 1, black thick line)*

Done.

*p8, li 10 - r could be reported in the figure next to the rmse, so it is available for each stations*

We have added the correlation factor to each plot.

*p8, li 17 - either write "high reflectivity" or "highly reflective"*

Done.

*p8, li 19 - (see methods section)*

Done.

*p8, li 23 - you give the uncertainty range here, how is it calculated?*

The uncertainty is calculated as the standard deviation. We have now specified this.

*p8, li 25 - Some of the models "do" have a steeper slope*

Done.

*p9, li 7 - You explain how the rms was calculated, the rmse value should also appear the text when discussing them.*

Done.

*p 12, li 7 - it is written that you don't include semi- or indirect cloud effects, or surface albedo modifications, how much approximately would the cloud effects and surface albedo modification impact the DAE?*

We have added the following: 'Jiao et al, 2015 estimates the Arctic surface radiative forcing from BC in snow to 0.18 Wm$^{-2}$, using deposited fields from the AeroCom Phase II models into an offline land and sea/ice model. There are few estimates of the semi-direct effects of aerosols, which is mostly due to BC. Bond et al. (2013) indicates a -0.1 Wm-2 global effect, equally split between direct and indirect effects, while a later study also indicates that the semi-direct effect

counteracts about 50% of the direct effect, independent of altitude (Samset and Myhre, 2015). None of these estimates are made specifically for the Arctic. Indirect cloud effects are likely different in the Arctic than at lower latitudes, in large because of the already bright surfaces in the Arctic. Also, cloud emissivity might be more important here, as thermal radiation dominates the dark winter months (Garrett et al, 2004).'

*p 12, li 17 - you give a percentage of the effect by doubling emission, what would be the corresponding value for the changes in lifetime? (also in the abstract)*

A doubling of the BC lifetime, result in a 39 % increase in Arctic AOD of BC. We have added this in the abstract.

*Figures:*

*Fig 3, caption: the average "observation" over the 9 stations*

This has been added.

*Fig 14, caption: "total" in small case*

Done.

*Fig 8: also here the multi model median could be added.*

Yes, originally we had the multi model median and the 25[th]/75[th] percentiles, but it was difficult to locate some of the other models, especially for the species with fewer models, so we decided to leave this out.

[revised manuscript text omitted]

| Model | Resolution | Meteorology | Mixing assumption, size distribution, and humidity growth factor |
|---|---|---|---|
| CAM4-Oslo | 2.5°×1.8°, 26 levels | GCM-generated | Internal vs. external mixing is determined on growth mechanism: coagulation, condensation, and cloud processing gives internal mixing with pre-existing particle. Maxwell-Garnett mixing for absorbing and transparent constituents, otherwise volume mixing. For internally mixed aerosols growth factor calculated from Kohler theory, taking hygroscopicity of each mixed constituent into account |
| HadGEM2 | 1.8°×1.2°, 38 levels | Nudged to ERA Interim data | External mixing. Size distributions prescribed for each aerosol component. Aitken, accumulation, coarse, and dissolved modes. Size distributions assumed lognormal for interaction with radiation. Hygroscopic growth is parametrized as a function of RH following Fitzgerald (1975) |
| ECHAM5-HAM | 1.8°×1.8°, 31 levels | Nudged to ECMWF analysis | 4 of total 7 modes are internally mixed; volume weighted mixing of refractive indices. Internal mixing for aerosol compositions within each mode, while external mixing is assumed among different aerosol modes. The humidity growth is based on Kappa-Koehler theory |
| OsloCTM2 | 2.8°×2.8°, 60 levels | ECMWF reanalysis | 8 bin sizes for SS and dust, log-normal size distributions in calculations of optical properties Hydrophilic BC is internal mixed; core shell type of mixing and an increase in absorption by 50%. The humidity growth is parameterized based on Fitzgerald (1975) |
| SPRINTARS | 1.1°×1.1°, 56 levels | Nudged to NCEP/NCAR reanalysis | 6 bins for dust, 4 bins for sea salt, 1 bin for sulfate, BC, and OA, with log-normal size distributions and particle growth as a function of relative humidity; 50% BC FF internally BC from other sources are internally mixed with POM. Other aerosols are externally mixed. The growth factors are according to Tang and Munkelwitz (1994) for sulfate, and Hobbs et al. (1997) for carbonaceous particles |
| GISS-MATRIX | 2.5°×2.0°, 40 levels | Nudged to NCEP winds | Mixing state is taken into consideration. Particles including BC have core shell structure; other particles use volume mixing approach. The size is prognostic and the mixing state assumption follows the population definitions in Bauer et al. (2008). Uptake of water calculated following the thermodynamical model EQSAM and for SS using the Lewis parameterization (Lewis and Schwartz 2004) |
| GISS-modelE | 2.5°×2.0°, 40 levels | Nudged to NCEP winds | Aerosol are externally mixed. Size distributions are prescribed. Sea salt, nitrate and sulfate get humidified following Lacis and Oinas 1991 and depends on ambient RH |
| CAM5.1 | 2.5°×1.8°, 30 levels | CAM5.1 | Internal mixing within each of 3 log-normal modes. Size varies with mass/number. Volume mixing of refractive indices of components within mode. Kappa Kohler theory using volume mean kappa. Dry if RH<RH_crystalization. Wet if RH>RH_deliquescence. Linear in RH between |
| BCC | 2.8°×2.8°, 26 levels | NCEP/NCAR reanalysis | Aerosols are externally mixed. The size spectrum of each aerosol is divided into 12 size bins. Kohler theory is used to calculate the humidity growth |
| GMI-MERRA-v3 | 2.5°×2.0°, 72 levels | Nudged to GEOS-5 MERRA reanalysis | External mixing. 5 bin sizes for dust, 4 bin sizes for seasalt, 3 bin size for nitrate and sulfate. All aerosols with log-normal size distributions. Based on Tang and Munkelwitz (1996) water activity formula for ammonium nitrate and ammonium sulfate. All others based on GADS OPAC |
| GEOS-Chem | 5.0°×4.0°, 47 levels | Nudged to GEOS-5 reanalysis | Optical properties calculated over 6 externally mixed species; inorganic ions (sulfate + nitrate + ammonium), OC (primary and secondary), BC, SS, and soil dust (4 size bins). 40 bins for secondary particles, 20 bins for sea salt, 15 bins for |

| | | | dust, 4 log-normal modes for BC and primary OC. A log-normal size distribution (except dust, gamma-distributions in the 4 size bins). The size distribution varies by hygroscopic growth |
|---|---|---|---|
| GOCART-v4 | 2.5°×2.0°, 30 levels | NASA GEOS-4 DAS reanalysis | External mixing. Parameterized with prescribed dry particle sizes: 8 bins for dust, 4 bins for sea salt, 1 bin for sulfate, BC, and OA, with log-normal distributions, particle growth parameterized as a function of RH. Humidity growth based on GADS (OPAC) |
| NCAR-CAM3.5 | 2.5°×1.9°, 26 levels | GCM-generated | Bulk-aerosol model, except 4-bins for SS and mineral dust |
| IMPACT | 5.0°×4.0°, 46 levels | DAO assimilation fields, reanalysis | 4 bin sizes for SS and mineral dust, 2 modes for pure sulfate with explicitly resolved size and coagulation and condensation of $SO_4$ with other aerosols |
| INCA | 3.8°×1.9°, 19 levels | ECMWF IFS reanalysis | 2 insoluble and 3 soluble modes with lognormal distribution. Size varies with number and mass affected by mixing, source and removal processes. Internal mixing is assumed with respect to removal by sedimentation and wet scavenging. Optical properties are calculated assuming external mixing. Optical properties assume mean size for each mode. 11 tabulated growth factors between 0 and 90% RH and 1 growth factor at 95% RH |
| TM5-V3 | 3.0°×2.0°, 34 | ECMWF ERA-Interim reanalysis | Five out of seven modes are internally mixed. Volume weighted mixing of refractive indices within each mode, using non-linear mixing rules. The median radius for each mode is taken to infer optical properties. The soluble particles are assumed to be in equilibrium with water vapor. Only sulfate and SS are influencing the water uptake |

[Figure]

**Figure S1: As in Fig 2, but with colors for the different models compared to AERONET. The black solid line is the AERONET mean, and the black dashed line is one standard deviation. R and rms is the correlation and root-mean-square between AERONET and the model median (model median is not shown in this plot).**

[Figure]

**Figure S2: As in Figure 3, but for JJA. The circles show the modelled AOD species JJA average for each station. The area of the circles is scaled to the model median total AOD.**

[Figure]

**Figure S3: Antarctic mean seasonal cycle of (total) BC AOD for simulations with GISS modelE (in colors) compared to the AeroCom models (in light grey). The darker grey AeroCom model is the GISS modelE AeroCom run. (a) shows emission perturbations for a doubling of BC emissions (fossil fuel and biofuel) in South Asia (green), East Asia (red), and Russia (yellow). (b) shows double (red) and half (green) of the e-folding time from hydrophobic to hydrophilic BC.**

---

## Author Response (AR2)

**Co-Editor Decision: Reconsider after minor revisions (Editor review) (02 Jul 2017) by Maria Kanakidou**

We would like to thank the editor again for the feedback and comments to improve the manuscript!

**Comments to the Author:**

*The discussion of the results has been improved in this revised version and most of the reviewer's comments have been satisfactorily addressed. However, there are more points that could further improved and are detailed below.*

*With regard to the reviewer #1 comment on the representativity of the year 2006 for the period 2000-2015. As a reply, I would expect, in addition to the authors reply, discussion on how 'typical' were the weather conditions including winds, temperatures, precipitation, as well as the fire emissions.*

In the Methods we have added the following: 'The AeroCom models have run simulations for year 2006 (with emissions for year 2000), while we have compared with observations from different available years 2000-2015. Comparing 2006 AOD values from CALIOP (only available July-Dec) with the 2007-2012 average, we find that 2006 is representative of the average (values varies 0%-36%).'

One year meteorology is of course not representative for the climatology, but in the AeroCom project the main focus was to run all the models for the same year. 2006 was not an 'untypical year', and 2006 was not an El Nino year. For the fire emissions, year 2000 was used as this was available at the time. The year 2000 does not stand out either (but 2006 does, however that was not the reason why 2000 was chosen). Since the models were run with 2006 meteorology, but with 2000 emissions, any comparison with observations must be taken with care.

*AOD calculations: It would be good to provide a supplementary table with the information on the optical properties used for different aerosol components.*

We agree that such a table would be very useful, but this is too much work load for this paper. Also, there is an overview paper about AeroCom Phase II in progress that will include this information.

*Page 6, line 3, please provide order of magnitude of the spread among models fund by Koffi et al. (2016).*

We have added: 'A comparison of the aerosol vertical extinction coefficient from 11 AeroCom models to CALIPSO has been performed in Koffi et al., (2016) showing that about half of the models capture the mean aerosol vertical distribution, but the bias depends highly on model, season, and region. They found that the models generally perform better over ocean than land (9 of 11 models reproduce the aerosol mean vertical distribution over ocean), while the models underestimate the mean aerosol distribution over land. This negative bias is especially pronounced in biomass burning and dust regions in Asia and Africa.'

*With regard to the overestimate of sea-salt and dust by GEOS-Chem and GOCART, there are published studies specific to these models and the representation of seasalt and dust with model evaluation, please use them to complement the discussion in page 8 (section 3.1)*

We have already added in page 11 that GOCART shows higher AOD values for dust compared to the other models, probably linked to an overestimation of dust emissions (Kim et al., 2014).

We have added: The high sea-salt values in Geos-Chem has been linked to an overestimation of sea-salt under high-wind conditions at mid- and high-latitudes (Jaegle et al. 2011). The model bias was reduced by adding a SST-dependent source function.

*Please add the reference of Stohl 2006 in the reference list.*

The reference is already there (page 21).

*Furthermore for the overall discussion, discussion of budget estimates in the polar regions would be very enlightening.*

Unfortunately, we do not have the aerosol mass from all the models available to discuss this.

*Page 8,line 22: please change 'AND' to small letters*

Done.

*Page 9, line 19 & 20: can you comment on the simulated decreasing trend of AOD? For instance, is this numerical or is it related to the emissions in the models or something else?*

The decreasing trend for the models lies within the model uncertainty and spread. The decreasing trend is dominated by a few models that has no specific similarities compared to the other models.

*Page 10, lines 28-29: can you comment on the nitrate simulations? And put these results in the context of the recently published in ACPD AEROCOM paper on nitrate intercomparison (Bian et al., acp-2017-359, 2017)*

We have added: 'For a first assessment of nitrate from multiple models compared with observations, see Bian et al. 2017, ACPD. Nine models from the AeroCom Phase III nitrate experiment show large diversity in their simulated nitrate concentrations, especially in remote regions. The authors link this spread to nitrate being involved in complicated chemistry, and that the nitrate concentrations depend on accurate simulations of precursors ($NH_3$, $HNO_3$, dust, and sea-salt).'

*Page 12, lines 24 & 25, decreases.*

Done.

*Page 13, line 22: please explain why.*

The relative spread for DRE is larger than the relative spread for AOD, and we have changed the manuscript accordingly. Thanks for pointing this out. The reason for the larger spread is linked to each model's radiation code (Stier et al. 2013).

[revised manuscript text omitted]

---

## Author Response (AR3)

**Co-Editor Decision: Reconsider after minor revisions (Editor review) (23 Aug 2017) by Maria Kanakidou**

*Comments to the Author:*

*Thank you very much for the additional revisions to your manuscript that are improving its quality. However, I would still miss 'numbers' to the remplies to :*

Thank you for the comments. We have changed the text following your suggestions:

*---Page 6, line 3 comment on the findings of Koffi et al - please provide numbers/order of magnitude/ percentage or absolute bias whatever is more convenient for you*

We have changed the text to: 'A comparison of the aerosol vertical extinction coefficient from 11 AeroCom models to CALIPSO has been performed in Koffi et al., (2016) showing that about half of the models capture the mean aerosol vertical distribution. The models generally perform better over ocean than land (9 of 11 models reproduce the aerosol mean vertical distribution over ocean), while the models underestimate the mean aerosol distribution over land. The annual mean multimodel mean absolute error is 11%, but the bias depends highly on model, season, and region. The negative bias is especially pronounced during spring and summer in source regions in Africa and Asia dominated by biomass burning (-17% to -26% bias) and dust (-8% to -23%).'

*---on your reply to the next comment, you mention 'The model bias was reduced by adding a SST-dependent source function' please provide number quantifying the improvement.*

We have changed the text: 'By adding a SST-dependent source function, the model bias was reduced from +64% to +33% for cruise measurements and from +32% to −5% for ground-based sites.'

*---Page 10, lines 28-29 comment - similar to above, please be more quantitative and provide numbers.*

We have added AOD values to the text.

[revised manuscript text omitted]